# Machine learning photodynamics decode multiple singlet fission channels in pentacene crystal

Zhendong Li[1,4], Federico J. Hernández [2,4], Christian Salguero[3], Steven A. Lopez [3] ✉, Rachel Crespo-Otero [2] ✉ & Jingbai Li [1] ✉

Crystalline pentacene is a model solid-state light-harvesting material because its quantum efficiencies exceed 100% via ultrafast singlet fission. The singlet fission mechanism in pentacene crystals is disputed due to insufficient electronic information in time-resolved experiments and intractable quantum mechanical calculations for simulating realistic crystal dynamics. Here we combine a multiscale multiconfigurational approach and machine learning photodynamics to understand competing singlet fission mechanisms in crystalline pentacene. Our simulations reveal coexisting charge-transfer-mediated and coherent mechanisms via the competing channels in the herringbone and parallel dimers. The predicted singlet fission time constants (61 and 33 fs) are in excellent agreement with experiments (78 and 35 fs). The trajectories highlight the essential role of intermolecular stretching between monomers in generating the multi-exciton state and explain the anisotropic phenomenon. The machine-learning-photodynamics resolved the elusive interplay between electronic structure and vibrational relations, enabling fully atomistic excited-state dynamics with multiconfigurational quantum mechanical quality for crystalline pentacene.

The discovery of singlet fission (SF) has triggered the rapid development of organic photovoltaic materials to achieve higher solar conversion efficiencies than those observed for conventional semiconductor solar cells[1–6]. SF is a spin-conserving process that converts a high-energy singlet exciton into two low-energy triplet excitons[7–9]. It provides an ideal tool to harvest the excess light energy higher than the band gap of solar cells. Many works have studied the SF process in a wide range of organic molecules, such as perylene[10], terrylenediimide dimer[11], diphenylisobenzofuran[12], quinoidal thiophenes[13,14], aza-cibalackrot[15], carotenoids[16], tetracene[17], pentacenes[18–20], and hexacene[21]. They showed that SF can take place in subpicoseconds, but the mechanistic origin of such an ultrafast process is not fully resolved. The lack of understanding of this fundamental process and missing mechanisms substantially limit progress toward new materials for SF-based devices. Studying the SF

mechanism will contribute to a deeper understanding of how to control the rate and quantum yields of SF in devices, helping maximize the energy efficiency of SF solar cells.

Crystalline pentacene is especially attractive because it generates triplet excitons in 80 fs[22]. The pentacene crystal contains five types of dimers (Fig. 1a), and the maximum electronic overlap is in the herringbone and parallel forms (Fig. 1b). Two decay time constants (78 and 35 fs) can be identified from the transient absorption (TA) spectrum of pentacene crystals[23], suggesting two distinct SF channels. The polarized TA microscopy showed the quantum decoherence rate along the parallel direction is 2.5-fold faster than along the herringbone direction[24], suggesting that SF in the parallel dimer is faster than in the herringbone dimer. A recent time-resolved photoemission study observed a mixed nature of local excitation and charge transfer in SF[25].

[1]Hoffmann Institute of Advanced Materials, Shenzhen Polytechnic University, Shenzhen 518055, People's Republic of China. [2]Department of Chemistry, University College London, London WC1H0AJ, UK. [3]Department of Chemistry and Chemical Biology, Northeastern University, Boston, MA 02115, USA. [4]These authors contributed equally: Zhendong Li, Federico J. Hernández. ✉e-mail: s.lopez@northeastern.edu; r.crespo-otero@ucl.ac.uk; lijingbai@szpu.edu.cn

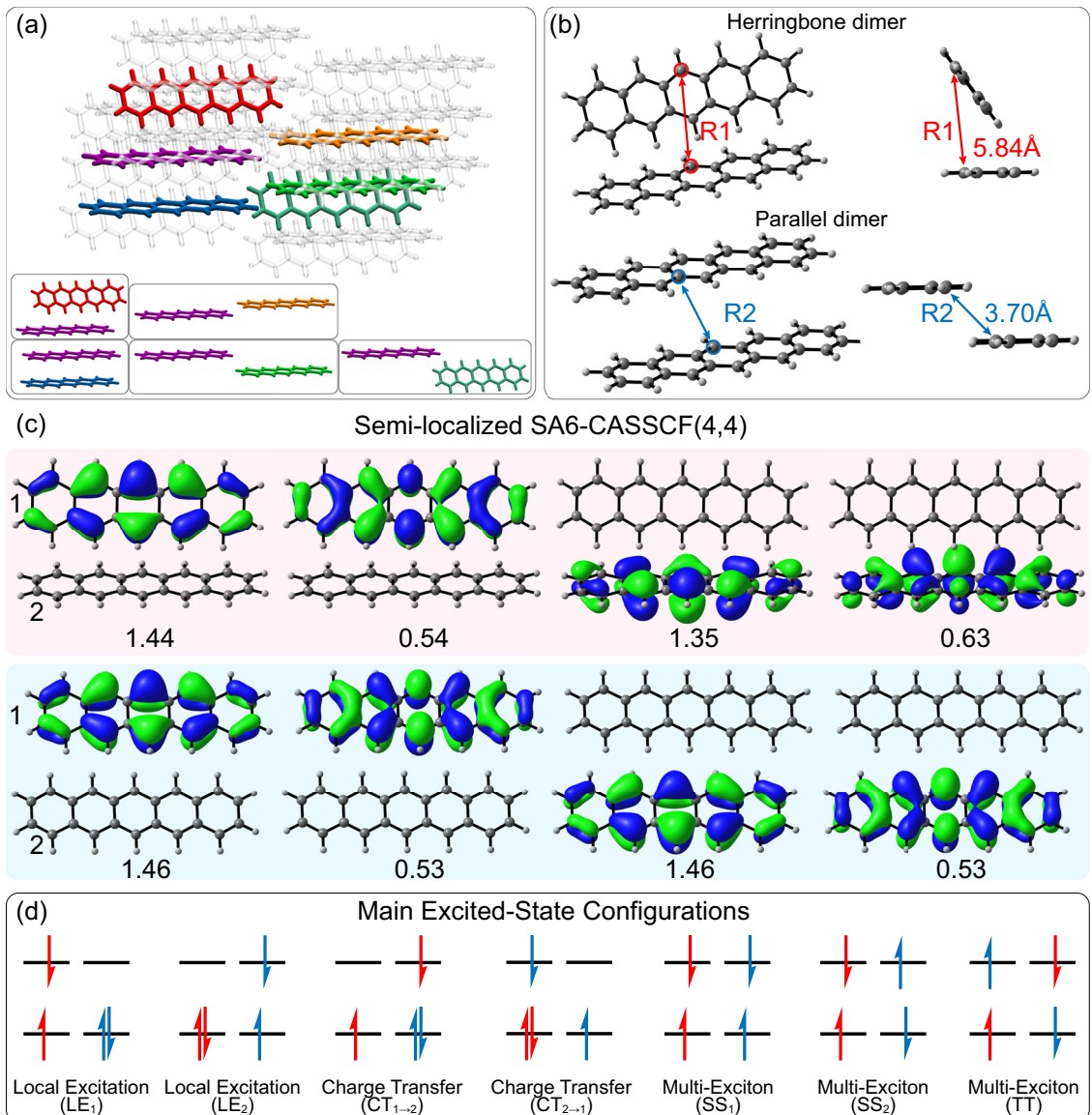

**Fig. 1 | Overview of pentacene dimer electronic structures. a** The crystal structure of pentacene with five types of dimers. **b** The geometries of the herringbone and parallel dimers optimized with ωB97XD/def2-TZVP. The intermolecular distances are defined by the carbon atoms in each central ring. **c** The semi-localized active space of the herringbone (*top*) and parallel (*bottom*) dimers, computed with the SA6-CASSCF(4,4)/cc-pVDZ method. The occupations are shown under the orbitals and averaged over 6 states. The number 1 and 2 denote the monomer 1 and 2. **d** Schematic representations for the main excited-state electronic configurations observed in the $S_1$ and $S_2$ states, computed with the SA6-CASSCF(4,4)/cc-pVDZ methods on the basis of the semi-localized active orbitals in panel (**c**). The subscripts indicate the monomer contributing to the electronic transitions.

Quantum chemical calculations by Deng et al. suggest that the anisotropic vibronic coupling of the pentacene tetramer in the crystal is responsible for the distinct SF channels[24]. However, the role of molecular vibrations is elusive, which has prevented a holistic understanding of the role of dimer morphology in controlling the SF rates. Besides the conventional SF generated from a bright $S_1$, the most recent TA experiments are consistent with the direct excitation to the dark spin-entangled triplet pair (TT) state[26].

Most theoretical studies have employed exciton models (i.e., Frenkel's model[27–29]) to explain the SF mechanism in the pentacene crystal[30–32]. Static quantum mechanical (QM) calculations based on the exciton models have revealed essential roles of the charge-transfer (CT) states[33,34], doubly excited (DE) states[35], and multi-exciton (ME) states[36,37]. The excited-state potential energy calculations suggest that increasing the intermolecular distances (Fig. 1b) changes the nature of the $S_1$ state of the pentacene dimer from CT to DE with ME character

and promotes SF[38]. This finding implies that the elongation of the intermolecular distance may help disentangle the elusive SF mechanism in the excited-state dynamics of the pentacene crystal. Exciton models have recently been implemented for the nonadiabatic dynamics simulations of these systems[39,40], but the high computational cost of the excited-state calculations for pentacene dimers in crystalline environments has prevented full-atomistic, on-the-fly nonadiabatic dynamics simulations from being combined with multiconfigurational calculations. Seiler et al. performed the Ehrenfest dynamics for the pentacene unit cell using time-dependent density functional theory (TDDFT) with no SF time constant information[41]. Wang et al.[42] simulated the excited dynamics of the pentacene dimers using the trajectories surface hopping method with classical path approximation, which predicted an SF time constant of 700 fs. Zheng et al.[43] and Peng et al.[44] performed the multi-configuration time-dependent Hartree method with selected vibrational modes,

respectively. Although the predicted SF time constants were improved to 70–120 fs, their results only showed an 80% SF yield. The discrepancy between the computations and experiments resulted from the lack of multiconfigurational calculations and full-dimensional nuclear dynamics.

We have overcome previous theoretical limitations with complete active space self-consistent field (CASSCF) calculations (Fig. 1c) to fully describe the electronic configuration interactions in the pentacene dimer (Fig. 1d). We trained neural networks (NNs) to accelerate the CASSCF calculations for computing the excited-state dynamics of pentacene dimers in crystals in the multiscale machine learning (ML) photodynamics simulations[45] in an electrostatic embedding ONIOM scheme[46]. Our simulations show two possible SF mechanisms via the CT-mediated and coherent photoexcitation of the herringbone and parallel dimers in the pentacene crystal, resulting in four unique SF channels. The predicted SF time constants are in excellent agreement with the experiments. The trajectories reveal the quasi-one-dimensional intermolecular stretching in the pentacene dimer during the SF process, which provides new insights for understanding the anisotropic SF phenomena in pentacene crystal.

## Results

### Pentacene crystal models

The pentacene crystal models comprise a photoexcited dimer inside a rigid crystal environment generated from the $3 \times 3 \times 3$ supercell. The herringbone (Fig. 2a) and parallel (Fig. 2b) dimers have 81 and 82 pentacene molecules in their rigid crystal environment, respectively. We compute the excitation energies of the dimers with electrostatic embedding (ee) six-state averaged (SA6) CASSCF(4,4)/cc-pVDZ calculations, where the active space is selected according to the frontier molecular orbitals of each pentacene (Supplementary Fig. 2). The restrained electrostatic potential (RESP) charges of surrounding molecules are embedded to account for the polarization from the crystal environments to the photoexcitation of pentacene dimers. The total energy combines the ee-SA6-CASSCF(4,4)/cc-pVDZ and GFN2-xTB calculations in a two-layer ONIOM scheme[46,47], and details are provided in the Methods section. Our benchmarks showed that the SA6-CASSCF(4,4)/cc-pVDZ method produced consistent electronic structures with XMS6-CASPT2(4,4)/ANO-S-VDZP results and excited state potential energies are in line with the mixed-reference spin-flip

(MRSF)-TDDFT[48] results (Supplementary Fig. 3-S5), in agreement with previous studies[36,49].

The simulated absorption bands (Fig. 2c and d) show zero intensities at most low-lying wavelengths of the adiabatic $S_1$ state (Supplementary Fig. 7a and b), due to the dominant TT character (Supplementary Fig. 8). They result in the lowest optical bright states in the adiabatic $S_2$ state with a local excitation (LE) character. The mixing of LE and TT with CT configurations in the $S_2$-Franck–Condon (FC) points suggests that $S_2$ could turn into a TT state via $S_2 \rightarrow S_1$ transitions (Supplementary Fig. 8). Thus, the photoexcitation to the adiabatic $S_2$ state informs the CT-mediated SF pathways. The adiabatic $S_1$ state displays minor transition-allowed regions with the help of the CT-mediated mixing of LE and TT characters (Supplementary Fig. 8). The photoexcitation in this region could directly generate the TT state, corresponding to the coherent SF pathways. The above results suggest a coexistence of the CT-mediated[25] and coherent[26] SF pathways reported in recent experiments. Therefore, we perform ML-photodynamics simulations[45] to study the SF mechanisms in both pathways.

### Photodynamics of pentacene dimer in crystals

We use ML-photodynamics simulations to accelerate the SA6-CASSCF(4,4)/cc-pVDZ calculations (Supplementary Table 1). The NN training data includes the energies and gradients of 6 singlet states of the dimers, including the ground state, computed with the ee-SA6-CASSCF(4,4)/cc-pVDZ calculations. We ignore the NACs between the non-adjacent states and approximate the NACs between adjacent states with the curvature-driven time-dependent couplings ($\kappa$TDC)[50,51], derived from the Baeck–An approximation[52]. A recent benchmark showed excellent agreement between $\kappa$TDC and ground-truth NACs, especially when the energy gap is small (<0.1 eV)[53,54]. Thus, we use the NN-predicted energies to compute the $\kappa$TDC in the ML-photodynamics simulations when the energy gap is <0.1 eV (Supplementary Fig. 9).

The initial training sets include 1000 Wigner-sampled structures of the pentacene dimers at the zero-point energy level. We expand the training sets with another 2000 structures by rescaling the atomic displacements in all vibrational modes to 90% and 80% with the Wigner sampling. A recent report showed this approach is effective in minimizing the NN errors for large molecules with complex molecular

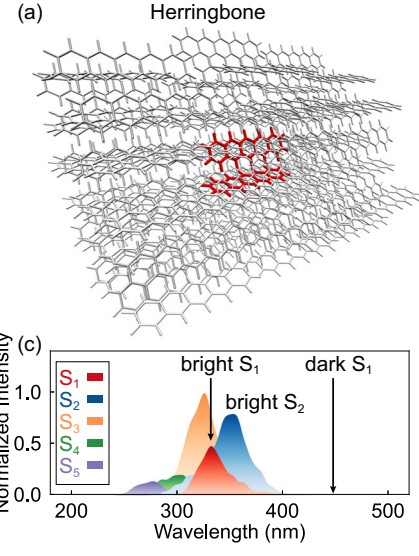

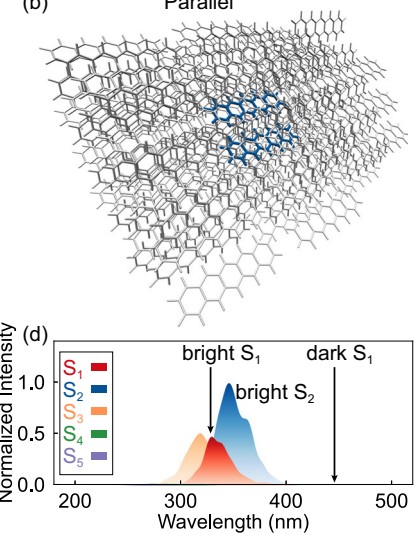

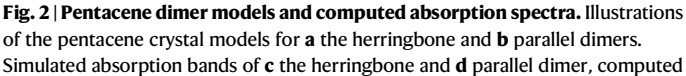

**Fig. 2 | Pentacene dimer models and computed absorption spectra.** Illustrations of the pentacene crystal models for **a** the herringbone and **b** parallel dimers. Simulated absorption bands of **c** the herringbone and **d** parallel dimer, computed at the SA6-CASSCF(4,4)/cc-pVDZ level. The intensities are normalized to the maximum average oscillator strengths. The terms bright and dark indicate the regions of transition-allowed and forbidden wavelengths.

structures[55]. The initial training sets are further amended with the adaptive sampling[56] to collect the undersampled structures in the ground- and excited-state potential energy surfaces. The final training sets increase to 4211 and 3455 data points for the herringbone and parallel dimers. Details of the adaptive sampling are provided in the Methods section.

We first launch the photodynamics simulations from the lowest optical bright state (i.e., the adiabatic state $S_2$), where the $S_2 \rightarrow S_1$ transition describes the CT-mediated SF process. The excited-state decay time constants (35 and 78 fs) measured in the TA spectrum of the pentacene crystal[23] suggest that 90% of the pentacene excited-state population arrives at the $S_1$ state in 81–180 fs. As such, we set the ML-photodynamics simulation time to 200 fs with a step size of 0.5 fs. The fewest switches surface hopping (FSSH)[57,58] with NN-predicted $\kappa$TDC is used to compute the non-adiabatic transition probabilities. We obtained over 500 trajectories for the herringbone and parallel dimers to obtain statistically sufficient data for investigating the SF mechanisms.

Figure 3a and b illustrate the state population dynamics of the pentacene dimers in the crystal. The $S_2$ relaxation undergoes the $S_2 \rightarrow S_1$ transition, where 95% of the herringbone dimers and 97% of the parallel dimers land on the $S_1$ state in 200 fs. The other trajectories remain in the $S_2$ or hop to the $S_3$ state. No trajectories are found in the $S_0$, $S_4$, or $S_5$ states. The $S_2$ populations fit an exponential decay time constant of 61 and 33 fs in the herringbone and parallel dimers.

We measure the intermolecular distances R1, R2, and the lateral motions Rz (Fig. 3c and d) to determine the structural changes in the trajectories. The $S_2$ relaxation of the herringbone dimer shows constant elongation of R1, where the average value increases from 5.84 Å at the $S_2$-FC points to 5.92 Å at the $S_2/S_1$ surface hopping points (i.e., the structures where $S_2 \rightarrow S_1$ transition occur) and 6.39 Å at the end of the dynamics (Fig. 3e). The average value of Rz near zero suggests the back-and-forth motions lead to no offset between the monomers (Fig. 3f). The parallel dimer undergoes a similar process. The average value of R2 in the parallel dimer increases from 3.75 Å at the $S_2$-FC points to 4.24 Å at the $S_2/S_1$ surface hopping points and reduces to 3.84 Å at the end of the dynamics (Fig. 3g). The average value of Rz is about zero (Fig. 3h). Recomputing the final

snapshots in the trajectories with the SA6-CASCI(4,4)/cc-pVDZ calculations confirmed the formation of the TT state at the end of simulations (Supplementary Table 4). Thus, these trajectories indicate two competing SF channels via the herringbone and parallel dimers in the pentacene crystal.

Based on the fitted time constants (Fig. 3a and b), the SF in the parallel dimer (33 fs) is faster than that in the herringbone dimer (61 fs), in line with the polarized TA microscopy experiments[24]. The predicted SF time constants also match with the decay time constants (35 and 78 fs) observed in the TA spectrum of pentacene crystal[23]. It suggests that the two excited-state decays observed in the TA spectrum[23] are attributed to two competing SF channels. The predicted time constant is 1.8 times longer in the herringbone dimer than in the parallel. This ratio is in excellent agreement with the ultrafast polarized transient absorption microscopy experiment, which reported a factor of 2.5 between the SF time constant measured in the herringbone and parallel direction[24]. Previous experiments also reported an anisotropic SF in the hexacene crystal with a factor of 4[21], where we anticipate a similar anisotropic SF mechanism to the pentacene.

We collected over 300 trajectories from the transition-allowed $S_1$-FC regions to determine the coherent SF mechanism. We noted a fast exchange of the $S_1$ and $S_2$ population in 12 fs, where 22% of the herringbone and 34% of the parallel dimer jump to $S_2$ and then quickly hop back to $S_1$ (Supplementary Fig. 10). 98% of the herringbone and 99% of the parallel dimers stayed in $S_1$ at the end of the simulations. The trajectories show similar intermolecular motions to the CT-mediated SF mechanism. The average trajectory of the herringbone dimer undergoes the elongation of the intermolecular distance R1 from 5.88 Å at the $S_1$-FC points to 6.36 Å at the final step (Fig. 3i) with slight absolute lateral displacement (Fig. 3j). The average trajectory of the parallel dimer shows an increased R2 from 3.75 to 4.23 Å during the dynamics (Fig. 3k), while the average value of Rz is almost unchanged (Fig. 3l). The SA6-CASCI(4,4)/cc-pVDZ calculations confirmed the dominant TT characters in the final snapshots of both dimers (Supplementary Table 4). These results suggest that coherent SF competes in two dimers.

To compare the intermolecular and intramolecular motions, we compute the distance matrix (DM) of the pentacene dimers in the

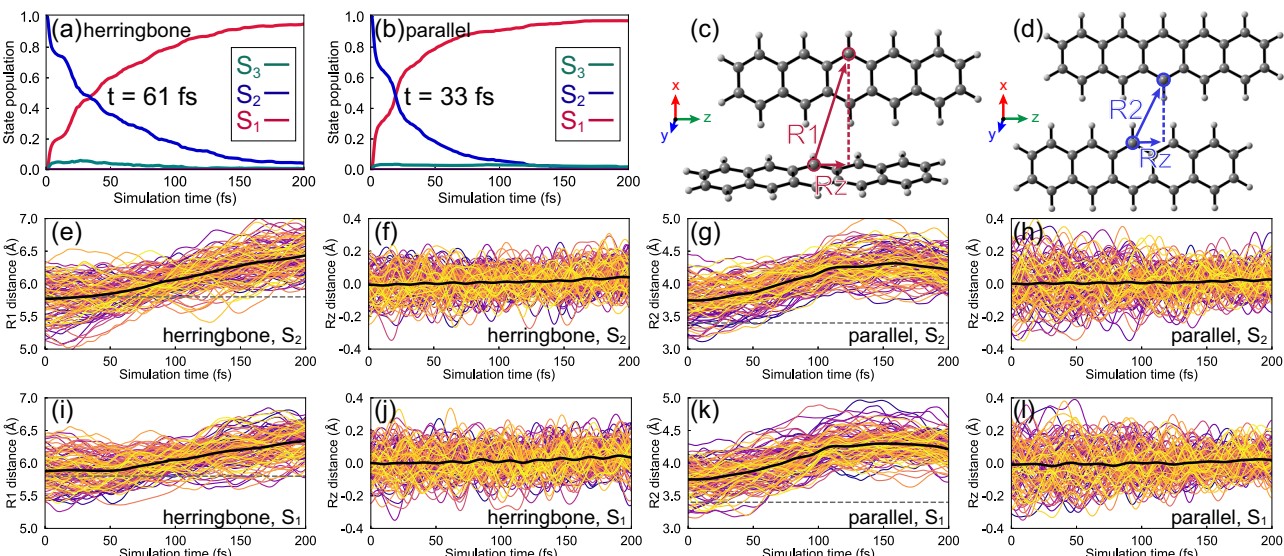

**Fig. 3 | State population dynamics and trajectory plots.** State population dynamics of **a** the herringbone and **b** parallel dimers in 200 fs ML-photodynamics simulations from the $S_2$-FC points. Definition of intermolecular distances R1, R2, and the projected lateral displacement Rz in **c** the herringbone and **d** parallel dimers. Plots for 100 randomly selected trajectories of **e** and **f** the herringbone and **g** and **h** parallel dimers started from the $S_2$-FC points. Plots for 100 randomly selected trajectories of **i** and **j** the herringbone and **k** and **l** parallel dimers started from the transition-allowed $S_1$-FC points. The gray dashed lines indicate the position of R1 = 5.8 Å in (**e**) and (**i**) and R2 = 3.4 Å in (**g**) and (**k**). The black curves show the average trajectory.

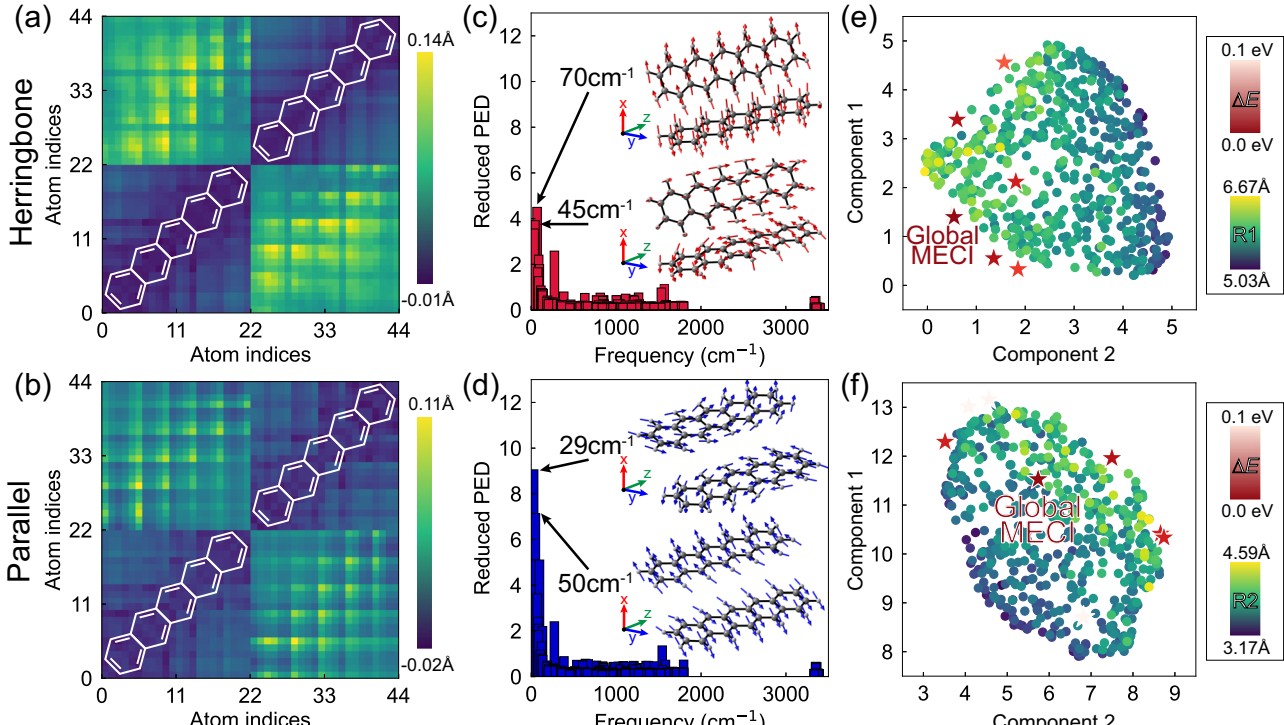

**Fig. 4 | Characterizations of pentacene dimer trajectories.** Differential distance matrices of **a** the herringbone and **b** parallel dimers based on the $S_2$-FC structures and the $S_2/S_1$ surface hopping structures. The distance matrices are defined by the intermolecular distances between the corresponding carbon atoms in the monomers. The atoms 1–22 and 23–44 refer to the carbon in monomers **1** and **2**, respectively. The diagonal blocks (*bottom-left* and *top-right*) describe the intramolecular distances, and the off-diagonal blocks (*bottom-right* and *top-left*) represent the intermolecular distances. Yellow corresponds to elongation; blue refers to shrink. Plots for the reduced potential energy distributions in the trajectories of **c** the herringbone and **d** parallel dimers with the two dominant $S_1$

vibrational modes. The 70 and 45 cm$^{-1}$ stretching of the herringbone dimer follows the *x* and *z*-axis; the 29 and 50 cm$^{-1}$ stretching of the parallel dimer follows the *y* and *z*-axis. UMAP clusterings of the $S_2/S_1$ surface hopping structures of **e** the herringbone and **f** parallel dimers. The hopping points are colored from green to yellow following the increasing order of the intermolecular distances. The red stars mark the locations of the optimized $S_2/S_1$ conical intersections, whereas the darkest red star represents the global minima of the conical intersections. The color bars illustrate the relative energies of the conical intersections and the intermolecular distances in the $S_2/S_1$ surface hopping points.

trajectories. The DM includes all pairwise distances between carbon atoms. Figure 4a and b illustrate the differential DMs between the $S_2$-FC and $S_2/S_1$ surface hopping structures for the herringbone and parallel dimers. The almost unchanged values in the diagonal blocks suggest little contribution from the intramolecular vibrations to the $S_2 \rightarrow S_1$ transition. In contrast, we find significant changes in the diagonal values of the off-diagonal blocks, indicating that SF in the pentacene crystal is mainly associated with the increasing intermolecular distance between the monomers. The differential DMs between the transition-allowed $S_1$-FC structures and the final snapshots in the trajectories demonstrate the same elongation of the intermolecular distances (Supplementary Fig. 11).

As the CT-mediated and coherent SF pathways show similar structural changes, we focused on the trajectories from the $S_2$-FC regions to understand the origin of the elongated intermolecular distances. We projected the nuclear displacements in the pentacene dimer trajectories to the $S_1$ vibrational mode coordinates to understand the elongation of the intermolecular distances in the pentacene dimers (Supplementary Figs. 12 and 13). The potential energy distributions (PED)[59] show notable vibronic-active low-frequency motions governing the excited-state dynamics in the pentacene crystal, which are at 70 and 45 cm$^{-1}$ in the herringbone (Fig. 4c) and 29 and 50 cm$^{-1}$ in the parallel dimers (Fig. 4d), respectively. The 70 and 50 cm$^{-1}$ modes are associated with hindered intermolecular rotations leading to the intermolecular stretching of the monomers in the quasi-orthogonal direction. These results are consistent with previous studies on the essential vibrational modes triggering the SF

in the herringbone dimer[49]. Moreover, the combination of the quasi-orthogonal 70 and 50 cm$^{-1}$ modes in the herringbone and parallel dimer matches the cross-axial low-frequency mode reported in the pentacene tetramer, which showed strong vibrational coherence with a 35 cm$^{-1}$ phonon facilitating the anisotropic SF in the pentacene crystal[24]. Our findings suggest that the quasi-orthogonal intermolecular stretching of the herringbone (70 cm$^{-1}$) and parallel dimer (50 cm$^{-1}$) produce the anisotropic SF phenomena in the pentacene crystal.

According to the wave-packet dynamics by Duan et al., the low-frequency intermolecular vibrations could facilitate the SF of the herringbone dimer by forming the intermolecular conical intersection (CI)[60]. A similar role of the intermolecular CI was also reported in a pentacene derivative[61] and other molecular aggregates[62]. We optimized the structures of the $S_2/S_1$ surface hopping points in the trajectories to understand how the crystal environments affect the intermolecular CIs of the pentacene dimers. Our calculations showed several degenerate intermolecular CIs in both herringbone and parallel dimers (Supplementary Fig. 14). Figure 4e and f visualize the UMAP of the $S_2/S_1$ hopping points with the optimized intermolecular CIs. The clustering of the $S_2/S_1$ surface hopping points resembles the state-crossing regions, where most intermolecular CIs are at the edge of the crossing regions. The global minimum CI of the herringbone dimer is associated with a notably shorter intermolecular distance ($R1 = 6.18$ Å) than the majority of the $S_2/S_1$ surface hopping points ($R1 = 5.92$ Å); the parallel dimer shows the global minimum CI ($R2 = 3.99$ Å) near the center of the $S_2/S_1$ surface

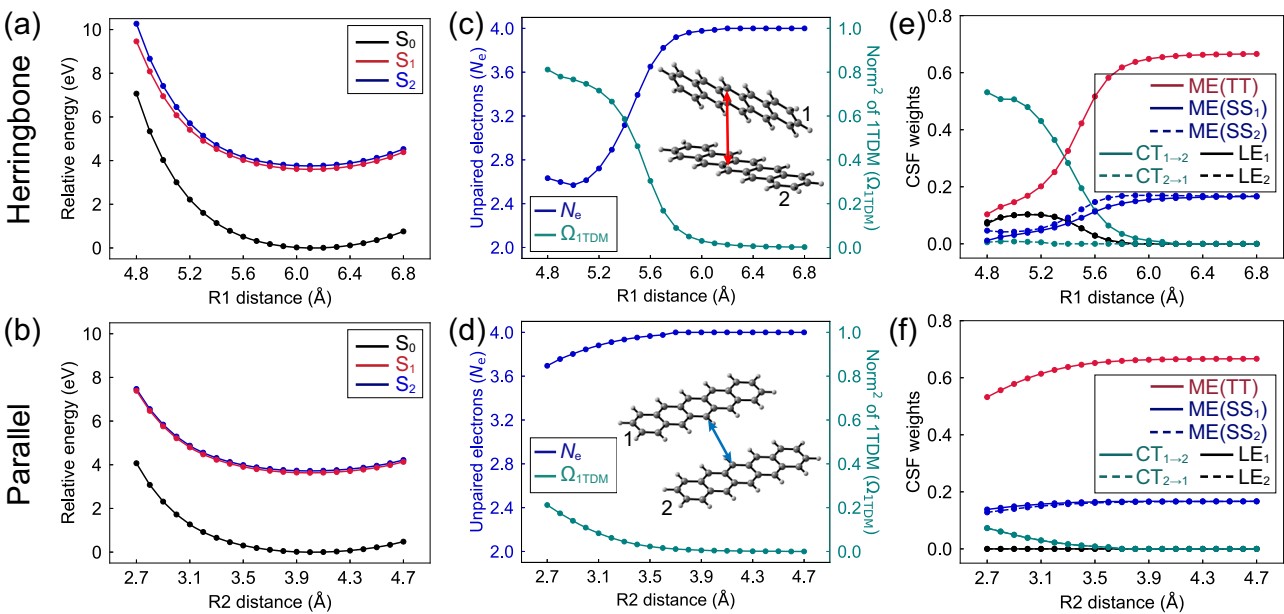

**Fig. 5 | Singlet fission mechanisms of pentacene dimers.** Plots for the potential energy curves of **a** the herringbone and **b** parallel dimers in the pentacene crystal, computed with the ee-ONIOM(SA6-CASCI(4,4)/cc-pVDZ:GFN2-xTB) method. The scan shows the ground-state local minimum of the herringbone and parallel dimers near 6.1 and 4.0 Å, respectively. These distances are slightly longer than the optimized values with the ωB97XD/def2-TZVP method due to the lack of dynamical correlation and dispersion corrections in the SA6-CASSCF(4,4)/cc-pVDZ calculations. Plots of the number of unpaired electrons ($N_e$) and squared norm of the one-electron transition density matrix ($\Omega_{1TDM}$) in the $S_1$ state of **c** the herringbone and **d** parallel dimers as functions of the intermolecular distances. $N_e = 2.0$ and $\Omega_{1TDM} = 1.0$ in a SE state, and $N_e = 4.0$ and $\Omega_{1TDM} = 0.0$ in a DE state. Plots of the weights of the configuration state wavefunction (CSF) of the local excitation (LE), charge transfer (CT), and multi-exciton (ME) in the $S_1$ state of **e** the herringbone and **f** parallel dimers as functions of the intermolecular distances. The subscripts denote the excitation sites. SS and TT refer to the singlet–singlet and triplet–triplet types of ME. The electronic configurations are omitted if their weights are <0.01.

hopping regions ($R2 = 3.84$ Å). Overall, the wide $S_2/S_1$ crossing regions are responsible for the efficient $S_2 \rightarrow S_1$ transitions in the pentacene crystal.

## SF mechanisms for the pentacene crystal

We performed rigid scans of the excited-state energies and electronic configurations to decode the intermolecular vibrational modes of pentacene dimers in generating the TT state. The vibrational modes contain the intermolecular elongations (i.e., $R1$ and $R2$), lateral motions, and rotations (Supplementary Figs. 15, 16a–d, 17a, b). However, the lateral motions and rotations do not control the excited-state characters of the pentacene dimer but increase the potential energies (Supplementary Figs. 18 and 19).

Figure 5a and b show the decreasing $S_1$ and $S_2$ energies as the elongation of $R1$ and $R2$ in the herringbone and parallel dimers. It explains the origin of the increasing intermolecular distances in the trajectories of both dimers in the CT-mediated and coherent SF pathways. The $S_1$ and $S_2$ energy plots show similar topology, with an average $S_2 - S_1$ gap of 0.24 and 0.08 eV in the herringbone (Fig. 5a) and parallel dimer (Fig. 5b). The broad regions of the close-lying $S_1$ and $S_2$ explain the wide $S_2/S_1$ crossing seam in our ML-photodynamics simulations. Their small gaps introduce a substantial mixing of CT, LE, and TT that facilitates the ultrafast $S_2$ and $S_1$ population transfer. We quantify the DE character in the $S_1$ state using the number of unpaired electrons ($N_e$) and the squared norm of the one-electron transition density matrix ($\Omega_{1TDM}$). The herringbone dimer shows $N_e$ of 2.6–2.7 and $\Omega_{1TDM}$ of 0.7–0.8 when $R1 < 5.2$ Å, indicating a dominant single exciton (SE) character in $S_1$ (Fig. 5c). The value of $N_e$ immediately increases to 3.9, and $\Omega_{1TDM}$ reduces to 0.09 at $R1 = 5.8$ Å, showing notable DE character in the $S_1$-FC point (5.84 Å). In the parallel dimer, $N_e$ increases from 3.7 to 4.0, and $\Omega_{1TDM}$ decreases from 0.21 to 0.03 when R2 approaches 3.4 Å (Fig. 5d). Thus, the $S_1$-FC point (R = 3.70 Å) of the parallel dimer is also a DE state.

The active orbitals are delocalized over the dimer in the potential energy curve calculations (Supplementary Fig. 2). Thus, the triplet configuration of each monomer cannot be explicitly described due to the orbital mixing between the monomers. As such, we performed the ee-SA6-CASCI(4,4)/cc-pVDZ calculations with the localized active orbitals[63] on the monomer (Fig. 1c). In Fig. 5e, the herringbone dimer shows more than 50% of the $CT_{1\rightarrow2}$ character from monomer **1** to **2** in the $S_1$ state when $R1 < 5.0$ Å. It also associates minor contributions (7%) from local excitations (LE) of monomers **1** and **2**. The population analysis predicts a $CT_{1\rightarrow2}$ of $0.44e$ at 5.1 Å (Supplementary Fig. 20b), which decreases to $0.1e$ near the $S_1$-FC geometry (5.84 Å). Continuously increasing $R1$ leads to a rise of the DE state with a TT character up to 67%, which confirms the TT-type DE state favoring the SF process. In addition, we find a competing ME character corresponding to two coupled SE states of both monomers, which increases from 0% to 17% along $R1$. In the parallel dimer, $S_1$ shows a weak SE character with $CT_{1\rightarrow2}$ and $CT_{2\rightarrow1}$ configurations, which decreases from 7% to 0% when $R2$ increases from 2.7 to 3.7 Å (Fig. 5f), in line with the absent CT in the charge analysis (Supplementary Fig. 20e). $S_1$ exhibits TT character from 53% to 67% when increasing $R2$, accompanied by 17% of the ME character resulting from the coupled SE character at each monomer. These findings agree with the previous gas-phase studies on the herringbone[37] and parallel[35] dimers, where $S_1$ changed from a CT-type SE to a DE state with increasing intermolecular distances.

Overall, our results show that the TT character of the pentacene dimers is controlled by quasi-one-dimensional intermolecular stretching along with $R1$ and $R2$. The $S_1$ of the herringbone and parallel dimers turn into the TT state when $R1 > 5.8$ Å and $R2 > 3.4$ Å, respectively. These findings explain the TT character in the trajectories because they exceeded the aforementioned distance thresholds at the end of the simulations, independent of the starting FC regions (i.e., $S_2$ or $S_1$). The complete $S_2 \rightarrow S_1$ transitions suggest a 100% CT-mediated SF yield in both herringbone and parallel dimers. Moreover, the herringbone dimer shows a wider range of $R1$ (4.9–6.6 Å) than $R2$ of the

parallel dimer (3.1–4.5 Å). They resulted in only 54% of the $S_1$-FC structures of the herringbone dimers being immediately accessible for SF, whereas the ratio for the parallel dimer is 92%. These results explain why the SF in the herringbone dimer is slower than the parallel dimer. The highly accessible SF channels at the $S_1$-FC regions explain the ultrafast SF process in the pentacene crystal.

## Discussion

We applied the multiscale ML-accelerated photodynamics approach to study the CT-mediated and coherent SF mechanism in the pentacene crystal. This approach integrated the neural networks trained with SA6-CASSCF(4,4)/cc-pVDZ data and the GFN2-xTB calculations, which allowed us to explore the excited-state dynamics in the pentacene crystal with multiconfigurational quality of theory in a full-atomistic manner. The unprecedented ML photodynamics trajectories provided statistically sufficient samples over a broad range of the excited-state conformational space of the pentacene dimer, presenting high-fidelity structural information to disentangle the elusive intra and intermolecular vibrations involved in the SF mechanism.

The multiconfigurational calculations confirmed the coexistence of the CT-mediated and coherent SF pathways, forming the TT state at the end of simulations. In each pathway, the trajectory analysis revealed two competing SF channels in the herringbone and parallel dimers. The $S_2$ lifetimes (61 and 33 fs) in the CT-mediated pathway are in excellent agreement with the TA experiments (78 and 35 fs). The analysis of the potential energy distributions in the trajectories uncovered two intermolecular stretching modes (70 and 50 cm$^{-1}$) that separated the monomers in the herringbone and parallel dimers in the crystal. Combining these two modes explains the formation of the cross-axial low-frequency vibration of the pentacene tetramer at 35 cm$^{-1}$, as reported in previous polarized TA microscopy experiments. The quasi-orthogonal directions of the intermolecular stretchings in the herringbone and parallel dimers also explain the anisotropic SF phenomenon in the pentacene crystal observed in the TA experiments.

The potential energy scans with ee-ONIOM(SA6-CASSCF(4,4)/cc-pVDZ:GFN2-xTB) calculations showed that the elongation of the intermolecular distances leads to rapid relaxation of the $S_1$ and $S_2$ states. These results confirmed that the quasi-one-dimensional intermolecular stretchings are the driving force behind the SF in the pentacene crystal. Evaluations of the unpaired electron numbers and the squared norm of the one-electron transition density matrix demonstrated the electronic nature of $S_1$ changes from a CT state to a TT state with increasing intermolecular distances in the herringbone dimer. Further analysis showed that only 52% of the $S_1$-FC structures of the herringbone dimer are immediately accessible for SF, whereas the ratio for the parallel dimer is 92%. These findings explain the faster SF in the parallel dimer than that in the herringbone dimer. The different SF rate constants in the herringbone and parallel dimers result in the anisotropic SF phenomenon in the pentacene crystal. Overall, the coexistence of the CT-mediated and coherent SF pathways in the herringbone and parallel dimers highlights multiple highly efficient SF channels in the pentacene crystal.

## Methods

### Multiscale quantum mechanical calculations

The experimental crystal structure of pentacene (CCDC:114447) was initially optimized using periodic DFT calculations along with the functional PBE-D2 as implemented in Quantum Espresso[64]. A Monkhorst–Pack $k$-point grid was chosen to match the unit cell parameters ($2 \times 2 \times 1$) and considered a basis set a cut-off of 60 Ry. Then, we generated the herringbone and parallel cluster models partitioning the crystal models into two layers: the dimer and the surrounding crystal shell, where the dimer includes 2 molecules and the crystal shell compresses 79 and 80 molecules for the herringbone and parallel dimers, respectively. The total energy was expressed using a two-layer

ONIOM scheme[46,47]:

$$E_{\text{total}} = E_{\text{GFN2-xTB, model}} - E^{\text{EE}}_{\text{GFN2-xTB, dimer}} + E^{\text{EE}}_{\text{QM, dimer}} \quad (1)$$

where the $E_{\text{GFN2-xTB,model}}$ term is the energy for the whole crystal model, computed with the GFN2-xTB method[65]. The $E^{\text{EE}}_{\text{GFN2-xTB,dimer}}$ and $E^{\text{EE}}_{\text{QM,dimer}}$ terms correspond to the electrostatic embedding GFN2-xTB and QM energies of the pentacene dimer, respectively. The polarizations from the crystal shell to the pentacene dimer were accounted for by embedding the RESP charges of the surrounding molecules into the GFN2-xTB and QM calculations. The gradients were obtained as the first-order derivatives of the total energy accordingly, where the nuclear positions in the crystal shells were frozen to describe the rigid environment in the lattice. In our ML-photodynamics simulations, the QM calculations were replaced by NN predictions, where all training data were computed with the electrostatic embedding of the RESP charges.

The pentacene dimer structures in the crystal were optimized using the ONIOM approach implemented in *fromage*[66], where the energies and gradients of the dimers and the RESP charges of the crystal shells were computed with the ωB97XD/def2-TZVP calculation using the Gaussian16 program[67]. In training data and the potential energy scan calculations, the pentacene dimers were computed with the SA6-CASSCF(4,4)/cc-pVDZ calculations using the BAGEL program[68] with the same RESP charges, where only the pentacene dimers are included to train NN. The active orbital localization in the SA6-CASCI(4,4)/cc-pVDZ calculations used the Pipek–Mezey method[62].

### Training data generation and NN training

The initial training data generation employed the Wigner sampling at the zero-point energy level to produce 1000 non-equilibrium geometries of the pentacene dimer according to the vibrational frequencies and modes computed with ωB97XD/def2-TZVP calculations. Another 2000 structures were obtained by rescaling the atomic displacements in all vibrational modes to 90% and 80% in the Wigner sampling. The training data were randomly split into training and validation sets in a 9:1 ratio.

We implemented a feed-forward neural network consisting of multiple perceptron layers based on the TensorFlow/Keras API for Python[69]. The NN computes the inverse distance matrix of the input molecule to predict the energies and gradients, where the atomic gradients are obtained from the analytical gradients of the NN. The NN employed a leaky softplus activation function. The loss function of the predicted energies and forces is combined with a ratio of 1:1 to ensure their physical relationship. The hyperparameters were optimized by a grid search over 384 NNs.

We used the adaptive sampling approach to explore the under-sampled data in the initial training set. The adaptive sampling propagates 100 trajectories from the $S_2$ state for 400 fs with a step size of 0.5 fs using a committee model of two independently trained NNs. We considered the standard deviation (STD) in the predicted energy and gradients of the NN committee as the uncertainty of the current prediction. The trajectories were stopped when the STD exceeded the empirical thresholds for energy (0.03 Hartree) or gradients (0.12 Bohr·Hartree$^{-1}$), respectively. The last geometries of the stopped trajectories were recomputed with the SA6-CASSCF(4,4)/cc-pVDZ calculations, including the charges of the crystal shell. The adaptive sampling retrained the committee model of NNs after adding the recomputed data to the initial training set. It then restarted the trajectories until the number of the out-of-sampled structures reached the minimum value. To speed up the adaptive sampling, the trajectories were propagated in the gas phase with only the charges of the crystal shell. The final training sets increased to 4211 and 3455 data points for the herringbone and parallel dimers. The mean absolute

errors in the final NN predicted energies were 0.0336–0.0363 and 0.0351–0.0421 eV for the herringbone and parallel dimers. The NN training, adaptive sampling, and ML photodynamics simulations use PyRAI²MD[43].

## ML-photodynamics simulations

We chose a rigid crystal environment in the ML-photodynamics simulations as the excited-state dynamics of the pentacene dimers were not affected by the flexibility of the crystal environment (Supplementary Table 5 and Supplementary Fig. 21). Detailed comparisons are available in Supporting Information.

The ML-photodynamics simulations propagated 1000 trajectories in the microcanonical ensemble (NVE) from the $S_2$-FC points of the pentacene dimers and 330 trajectories from the transition-allowed $S_1$-FC points in 200 fs with a step size of 0.5 fs. The probability of a nonadiabatic electronic transition was computed with Tully's fewest switches surface hopping (FSSH) algorithm[57,58], where we used the curvature-approximated time-derivative coupling (kTDC) method[50,51] to evaluate the NACs based on the NN predicted energy gaps. The kTDC method showed a good accuracy to the ground-truth NAC obtained with QM calculations when the energy gap was sufficiently small (e.g., 0.5 eV)[52,54]. Our tested QM photodynamics simulations using the NACs computed at the SA6-CASSCF(4,4)/cc-pVDZ level show the majority of the $S_2/S_1$ surface hops occurred with an energy gap <0.1 eV. Thus, we chose a threshold of 0.1 eV for computing the kTDC in our ML-photodynamics simulations.

The close-lying $S_2$ and $S_1$ energies made NN training difficult because their energy gaps could be one order of magnitude smaller than the NN prediction errors at the $S_2/S_1$ surface hopping regions. Thus, the errors in the NN-predicted energies could lead to artifacts in the potential energy curvatures, resulting in incorrect state population transfers in our ML photodynamics simulations. Our trajectory analysis removed the trajectories with incorrect state populations, e.g., exceeding 0–1. As a result, we obtained 571 and 544 trajectories from $S_2$-FC points and 292 and 318 trajectories from the transition-allowed $S_1$-FC points for the herringbone and parallel dimers, respectively.

We found the GFN2-xTB and GFN-FF[70] methods produced similar results in the ML-photodynamics simulations (Supplementary Fig. 21). A single ML-photodynamics trajectory computed with the ee-ONIOM(SA6-CASSCF(4,4)/cc-pVDZ:GFN2-xTB) requires 110 days using 6 CPUs. The ee-ONIOM(NN/GFN2-xTB) and ee-ONIOM(NN/GFN-FF) calculations finished in 4.6 days and 0.5 h, corresponding to a 24-fold and 5099-fold acceleration.

We projected the time-resolved nuclear displacements in the trajectories onto the $S_1$-state normal modes coordinates to evaluate the reduced potential energy distribution (PED) as follows:

$$Q_i(t) = \sum_A m_A \Delta r_A(t) \cdot v_{Ai} \tag{2}$$

where $v_{Ai}$ are the eigenvector matrix elements, $\Delta r_A(t)$ are the nuclear displacements, $m_A$ is the atomic mass, and $Q_i$ are the coordinates in the normal modes basis set. The potential energy in the $i$th mode is therefore calculated as

$$V_i(t) = (2\pi c \underline{v}_i Q_i(t))^2 \tag{3}$$

where $\underline{v}_i$ is the normal mode's wavenumber and $c$ is the speed of light. The potential energy per mode is then integrated along the trajectory. Finally, the PED is averaged across all selected trajectories and divided by $\underline{v}_i$ to yield the unitless reduced PED.

## Data availability

The RESP point charges, NN models, a small set of initial conditions and input files are available at https://github.com/mlcclab/PyRAI2MD_ publications/tree/main/Pentacene_dimers. Full training data, initial conditions and all trajectory data are available in Figshare, https://doi. org/10.6084/m9.figshare.28082003. Source data are provided with this paper.

## Code availability

The fromage code is available at https://github.com/Crespo-Otero-group/fromage. The PyRAI²MD code is available at https://github.com/mlcclab/PyRAI2MD-hiam. https://doi.org/10.5281/zenodo.14546617.

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

## Acknowledgements

J.L. acknowledges the funding support by National Natural Science Foundation of China Grants (22303053) and the Shenzhen Polytechnic University Research Startup Program (6023312036K). Z.L. and J.L. thank the support of the high-performance computing resources at Hoffmann Institute of Advanced Materials at Shenzhen Polytechnic University. Z.L. and J.L. gratefully acknowledge HZWTECH for providing computation facilities. S.A.L. acknowledges the NSF (NSF-CHE-2144556) for funding. C.S., J.L., and S.A.L. appreciate the assistance from the Northeastern Research Computing Team and the computing resources provided by the Massachusetts Life Science Center Grant (G00006360). R.C.O. and F.J.H. acknowledge funding from the Leverhulme Trust (RPG-2019-122). R.C.O. also thanks to EPSRC (EP/R029385/1) and UKRI (EP/X020908/2). R.C.O., F.J.H., and S.A.L. acknowledge the Royal Society for funding through an International Exchange Grant (IES\R2\222057).

## Author contributions

S.A.L., R.C.O., and J.L. conceptualized the project. Z.L. performed the DFT calculations under the supervision of J.L. F.J.H. and C.S. performed the TDDFT and CASSCF calculations under the supervision of R.C.O. and S.A.L. J.L. performed the ML photodynamics simulations. Z.L., F.J.H., C.S., and J.L. analyzed the data. J.L., F.J.H., and Z.L. prepared the manuscript. All authors revised the manuscript.

## Competing interests

The authors declare no competing interests.
