## [Peer review file · Nature Communications]

Machine learning photodynamics decode multiple singlet fission channels in pentacene crystal

Corresponding Author: Dr Jingbai Li

Version 0:

Reviewer comments:

Reviewer #1

(Remarks to the Author)

The manuscript is devoted to a computational investigation of singlet fission dynamics in a pentacene crystal. In particular, two pentacene dimers are separately considered, with herringbone and parallel orientations, inserted in an environment apt to simulate the interaction with the other pentacene molecules in the crystal. The ab initio on the fly nonadiabatic molecular dynamics simulations are accelerated by a machine learning approach with neural networks. For both dimers, the excited state dynamics leading to the TT state is ultrafast, with time constants in agreement with the experimental results. Overall, the work presented is impressive. Here are my remarks.

1) The title of the manuscript is very misleading. In particular, the authors claim that the approach considered in the present work is "beyond the Frenkel exciton model". Then, the reader expects that some sort of "divide and conquer" strategy is adopted, like in the excitonic approach, but including in some way also charge transfer states etc.. On the contrary, the method considered here is the standard "supermolecule" approach, which includes by construction all the configurations needed, but which is also limited to a very reduced number of chromophores (two, in the present case). My suggestion is therefore to change the title.

2) The decay dynamics is analyzed in terms of low frequency vibrational modes of the dimers. I am wondering how realistic is this analysis, taking into account that the vibrational modes considered have been obtained in a rigid crystal environment (if I understand correctly), and the low frequency modes could be highly perturbed if all the pentacene monomers of the cluster were allowed to move.

3) The lack of dynamical correlation in the casscf(4,4) calculations could affect the energetic order of the states, in particular favouring the quintet state with respect to the singlets. The authors should briefly comment on this.

4) In the simulated absorption spectra reported in figure S1, the band associated to the S2 state is reported to be at lower energy with respect to the band associated to the S1 state. Why? My understanding is that it is implicitly assumed that the states labeled as S1, S2, S3, etc. are adiabatic singlet states in energetic order. Moreover, the bright state in figure S1 appears to be the lowest excited singlet state (in particular for the parallel dimer), which does not match with what is said in the main text. Am I missing something?

5) A possible typo: The acronym CASSCAF, appearing several times in the manuscript

and in the SI, should be replaced by CASSCF? If not, please specify.

Reviewer #2

(Remarks to the Author)

This manuscript reports a computational study of singlet fission in pentacene crystals. The authors applied multiscale multiconfigurational approach, combined with machine learning, to simulate the dynamics of singlet fission in pentacene crystals. This is a nice, albeit incremental, improvement in our understanding of singlet fission in this most widely studied model system. The claim that "singlet fission mechanism in pentacene crystals is unresolved" is overstated. Extensive work in the last 15 years, both experimental and computational, in pentacene crystals and model dimers, have established perhaps the clearest mechanistic picture on singlet fission in this model system. The role of vibrational motions, particularly inter-molecular motions, has been extensively studied. Moreover, the sub 100 fs dynamics determined in experiments have also reproduced in many computational/simulation work, including a number of reports based on multiconfigurational interactions.

I would agree that the current work may have computational advantages in adding to this mechanistic picture, but it is by no means the final say. The model adopted by the authors, i.e., a pentacene dimer embedded in a rigid matrix, is also an approximation. Exciton delocalization in pentacene crystals is likely more extensive than what is captured by the dimer (see, e.g., J. Phys. Chem. Lett. 2013, 4, 2197). Such delocalization is reflected in the proposals of CT states in singlet fission, culminating the recent experimental verification (Nature 2023, 616, 275, ref. 25). The sampling of some low frequency modes, particularly the inter-molecular motion, in assisting conical interception type of dynamics, has its own limitations. The actual involvement of vibrational motions may be highly multidimensional. In this sense, the claim of novelty and importance is a stretch. I am not an expert on machine learning, but the authors' approach likely improves the computational efficiency. Report of such improvements belongs to a technical journal, not the broader readership of Nat. Comm.

Reviewer #3

(Remarks to the Author)

The work "Machine learning photodynamics beyond the Frenkel exciton model: Intermolecular vibrations trigger ultrafast singlet fission in pentacene crystal" by Li et. al. describes a novel and interesting way to simulate the photodynamics of a pentacene crystal using trajectory surface hopping without the use of the Frenkel exciton model.

Instead a multiscale cluster approach was used in combination with multiconfigurational CASSCF calculations of the relevant pentacene dimers (herringbone and parallel dimer).

To speed up the computations and to enable trajectory surface hopping simulations in a reasonable time frame, neural networks were employed to replace the (expensive) CASSCF calculations.

The resulting decay rates are in excellent agreement with the experimental values for both the herringbone and the parallel dimers.

The manuscript is generally well written and but the reviewer suggests its publication in Nature Communications only after the following questions/comments are addressed.

1. The authors write "This method was previously benchmarked against multireference methods with second-order perturbative corrections, which showed essentially the same topology in the excited-state potential energy surfaces (PES) for the pentacene dimer."

Here they cite Ref. 45 "Mechanism for Singlet Fission in Pentacene and Tetracene: From Single Exciton to Two Triplets" from Zimmerman et. al.

However, inside this paper, I could not find calculations with a second-order perturbation theory method on the pentacene dimer, neither using CASPT2 nor multi-reference perturbation theory (MRMP).

Instead the computationally efficient restricted active space double spin-flip (RAS-2SF) method was compared with CASSCF (see SI of Ref. 45) and both shifted to the MRMP(12,12) values of the pentacene monomers at the dissociation limit.

The authors of Ref. 45 write, that they compare with CASSCF but do not further specify the computational method. In a cited former publication of some of the same authors (<https://doi.org/10.1038/nchem.694>)

"Singlet fission in pentacene through multi-exciton quantum states" it is stated that for the PE scans CASSCF(8,8) was used for the dimer.

Therefore, I assume this was also used for the scan in the SI of Ref. 45.

Where was the benchmark with respect to the multireference methods with second-order perturbation corrections done, as I could not find it in Ref. 45?

Was the choice of CASSCF(4,4) based on the RAS-(4,4)-2SF method in Ref. 45?

If so, how do the two methods compare? And how well is it compared to the CASSCF(8,8) used in Ref. 20 of Ref. 45?

As the authors apply a machine learning (ML) approach to generate the CASSCF energies, why did they not use a higher level of theory like CASPT2?

Along these lines, how much more expensive would be the CASPT2 calculations?

In Ref. 20 of Ref. 45 it is stated that due to the lack of dynamic correlation in CASSCF an overestimation of the energy of the S1 occurs and

it is needed to shift the energy of the electronic states accordingly to the MRMP(12,12) results. Do the authors see a similar behavior in their calculation?

As the hopping rate is strongly effected by the energy gap between the relevant electronic states, are all excited states simply shifted by a constant factor

between the CASSCF and the multireference methods with second-order perturbative corrections, or are there changes in the energy gap between the two methods?

2. One difficulty in learning QM/MM data is to learn the influence of the MM sites on the QM structure, as there are many more configurations to consider. Are the MM (xTB) atoms included in the NN fit or is a pure ONIOM approach used, where only the QM atoms are included?

3. The authors state, that a single ML surface hopping dynamics was completed in 5 days due to the computational bottleneck of the GFN2-xTB calculations.

Is the use of GFN2-xTB necessary? Especially, when the ML approach is used or did the authors consider more approximate but computational cheaper schemes like the GFN-FF approach?

4. For the state analysis and to better characterize the final state of the trajectory, did the authors recompute (a selection of) the final structures obtained

with the ML learning surface hopping approach and perform ee-ONIOM(SA6-CASSCF(4,4) or CASCI/cc-pVDZ:GFN2-xTB) calculations on them? Or was the analysis purely done based on structural information (e.g. the R1/R2 values)?

If only structural information was used, it would be good to verify at least for a selection of the final structures of the dynamics that the final state (S1) is indeed a triplet state.

This point is crucial, as the singlet fission rate is defined as the S2->S1 decay rate.

5. The authors write that the average value of R1 is 5.92 Angstrom at the S2/S1 surface hopping points in herringbone dimers.

What are these "surface hopping points"? Are these structures at which a hop occurs?

6. The authors write that the ML-photodynamics simulations "are transferable to understanding the SF process in the crystals of pentacene and hexacene."

However, in the manuscript only pentacene results are shown. Is the statement regarding hexacene a (valid) guess or did the authors perform hexacene calculations as well?

7. For the neural network calculations, both energies and forces are predicted based on QM data.

One issue, when energies and forces are both fitted by the neural network is that the gradients are not the analytic derivatives of the energies, which can lead to issues during the dynamics (e.g. non conserved total energies).

Did the authors encounter this problem? One way to prevent this is to only fit the energies using the NN and compute the gradient analytically from the NN using an automated gradient approach (e.g. autograd of PyTORCH).

Did the authors also try this approach?

8. How exactly are the distances R1 and R2 defined? The authors write: "The intermolecular distances are defined by the carbon atoms in each central ring."

I assume, the definition is the one in Figure S3 of the supplementary material, but from the text and Figure 1, it would not be clear to me. It might be helpful to point to Fig. S3 a)/d) just for clarification.

This definition of the distances raises the question, how the displacements were done for the scans in Figure 4.

I assume, the pentacene molecules were displaced along the R1/R2 vector, while keeping all the other coordinates frozen? This in fact represents a translation between the two pentacene molecules.

The authors show nicely, how an increase in R1/R2 distance change the state character of the S1 to a triplet state, however, during the dynamics (Figure 2), how much of the changes of R1/R2 are due to this translation and how much is due to other coordinate

changes e.g. rotations especially along the Z-Axis (as defined in Figure 3 c/d) through the center of mass of the individual pentacene molecule?

In fact in Figure 3, the authors identify two low lying intermolecular rotations as the main vibronic active low-frequency motions.

Could the authors comment on the change in excited state character of the system along these rotation?

Minor remarks:

page 4: I assume, that the SA6-CASSCAF(4,4) is a typo and should be SA6-CASSCF(4,4), if not please explain the difference between these methods.

If yes, please change this also in the other cases.

page 8: "Since the trajectories showed efficient S2/S1 surface hoppings to continue the SF process"

Better use something like: "population transfer from S2 to S1" instead of "S2/S1 surface hoppings", if these is meant.

The link in the SI (https://github.com/mlcclab/PyRAI2MD_publications/Pentacene) does not work and should be updated

Version 1:

Reviewer comments:

Reviewer #1

(Remarks to the Author)

After the extensive clarifications and modifications provided by the authors, the manuscript is suitable for publication.

(Remarks on code availability)

Reviewer #3

(Remarks to the Author)

The work "Machine learning photodynamics decode multiple singlet fission channels in pentacene crystal" by Li et. al. describes a novel and interesting way to simulate the photodynamics of a pentacene crystal using trajectory surface hopping with a multiscale cluster approach. To speed up the computations and to enable trajectory surface hopping simulations in a reasonable time frame, neural networks were employed to replace the (expensive) CASSCF calculations.

The resulting decay rates are in excellent agreement with the experimental values for both the herringbone and the parallel dimer. The manuscript is generally well written and in the new version of the paper most of my initial questions and comments were answered. Still I have a few (new) open questions/comments that should be addressed before publication.

1. The authors clarified that they selected SA6-CASSCF(4,4)/cc-pvdz as the method of choice, that shows a good compromise of efficiency and accuracy for the NN data. However, I do not understand the argument how SA6-CASSCF(4,4) was chosen.

In the benchmark study SA6-CASSCF(4,4) is compared with SA6-CASSCF(8,8), XMS6-CASPT2(4,4) and MRSF-TDDFT. On page 4 the authors write: "Our benchmarks showed that the excited-state potential energy curves at the SA6-CASSCF(4,4)/cc-pVDZ level of theory are consistent with those obtained using XMS-CASPT2(4,4)/cc-pVDZ". But in both the answer to the reviewer as well as in the SI, the authors choose MRSF-TDDFT as the reference for the benchmark, because XMS6-CASPT2(4,4) "shows inconsistent results suggest that the XMS6-CASPT2(4,4)/cc-pVDZ method cannot capture all essential dynamical correlations". If this is the case, why do the authors emphasize that the SA6-CASSCF(4,4) is consistent with the XMS6-CASPT2(4,4) results (which I disagree, as it shows a different behavior for the two most important excited states)?

Following up on this, if MRSF-TDDFT was chosen as the reference for the benchmark and given that its computational costs is roughly half that of SA6-CASSCF(4,4), why was it not chosen as the method of choice?

The authors state in the SI that "The efficiency of the MRSF-TDDFT(PBE0)/cc-pVDZ calculations is realized by the newly designed quantum chemistry code, OpenQP, which is still under development and is only available for benchmark purposes while we finish this work".

However, I do not fully understand this statement. What do the authors mean by "is only available for benchmark purposes"? Was the method only available, after the molecular dynamics simulations were performed?

2. The authors compute the training data for the ML using an electrostatic embedding approach, where one pentacene dimer is the QM region and all the others are included as point charges.

The obtained NN model is used in an ONIOM approach, where the low level method is GNF2-xTB (GNF-FF).

In ONIOM the interaction between the active side (here the pentacene dimer/NN) and the environment (here the other pentacene molecules) is computed at the low level method.

From this it seems to me that the electrostatic interaction between the active side and the environment is (in parts) counted twice.

Is this accounted for in the ONIOM scheme in combination with ML/NNs?

3. The authors write at several points that they use SA6-CASCI(4,4) instead of SA6-CASSCF(4,4).

Is the same method meant? If yes, please use SA6-CASSCF in all places, if not why was SA6-CASCI used and how does it perform in comparison?

Minor remarks:

page 6: "The predicted SF time constants also match with the decay time constants (35 and 75 fs)". I assume that is a typo and it should be 78 fs.

SI:

page 1: "electron-hele distances"

(Remarks on code availability)

Version 2:

Reviewer comments:

Reviewer #3

(Remarks to the Author)

Thanks a lot for the extensive clarifications. In my opinion the manuscript is suitable for publication.

(Remarks on code availability)

Response to referees

Our answers are in blue, and changes are in red.

Reviewer #1 (Remarks to the Author):

The manuscript is devoted to a computational investigation of singlet fission dynamics in a pentacene crystal. In particular, two pentacene dimers are separately considered, with herringbone and parallel orientations, inserted in an environment apt to simulate the interaction with the other pentacene molecules in the crystal. The ab initio on the fly nonadiabatic molecular dynamics simulations are accelerated by a machine learning approach with neural networks. For both dimers, the excited state dynamics leading to the TT state is ultrafast, with time constants in agreement with the experimental results. Overall, the work presented is impressive. Here are my remarks.

1) The title of the manuscript is very misleading. In particular, the authors claim that the approach considered in the present work is "beyond the Frenkel exciton model". Then, the reader expects that some sort of "divide and conquer" strategy is adopted, like in the excitonic approach, but including in some way also charge transfer states etc.. On the contrary, the method considered here is the standard "supermolecule" approach, which includes by construction all the configurations needed, but which is also limited to a very reduced number of chromophores (two, in the present case). My suggestion is therefore to change the title.

- Answer: Thank you for your thoughtful suggestion on improving our title. We removed the "beyond Frenkel model" to avoid misinterpreting the scope of this work. We changed the title to "Machine learning photodynamics decode multiple singlet fission channels in pentacene crystal", which best summarizes the novelty of this work: the ML photodynamics revealed the fundamental intermolecular motions that trigger anisotropic SF channels in the pentacene crystal with two possible singlet fission (SF) mechanisms: 1) ultrafast charge-transfer-mediated internal conversion from the bright singlet state to dark triplet pair state or 2) direct coherent photoexcitation to the dark triplet pair state.

2) The decay dynamics is analyzed in terms of low frequency vibrational modes of the dimers. I am wondering how realistic this analysis is, taking into account that the vibrational modes considered have been obtained in a rigid crystal environment (if I understand correctly), and the low frequency modes could be highly perturbed if all the pentacene monomers of the cluster were allowed to move.

- Answer: Thank you for the comments on our rigid crystal model. We chose the rigid crystal environment for two reasons: 1) the pentacene dimers have sufficient volume to accommodate the excited-state vibrations evolved in singlet fission; 2) the singlet fission is ultrafast and is completed before the pentacene dimer responds to perturbations from the surrounding pentacene molecules.

We computed the Voronoi volume and the van der Waals (vdW) volume for each pentacene monomer to estimate the maximum available space inside the crystal and the occupied space, respectively. The ratio between Voronoi and van der Waals volume, called the volume index (V_i), shows the extent of the molecular flexibility. The V_i of the optimized structure in the herringbone and parallel dimer is 1.43 and 1.42, respectively. These values suggest large volumes available for the excited-state vibrations. Moreover, we performed the ML photodynamics simulations with a flexible crystal environment to assess the influence of the excited-state vibrations and dynamics. We chose the GFN-FF for computing the crystal environment because of its similar performance to GFN2-xTB and high efficiency. Please see our answer to Reviewer 3's Comment 3 for more details on the methods. The table below collects the initial and final average Voronoi, vdW volume, and V_i of both dimers in the ML photodynamics simulations with rigid (NN/GFN2-xTB, NN/GFN-FF) and flexible (NN/GFN-FF) crystal environment. The values in parathesis indicate the standard deviations.

Dimers	Model	Voronoi (\AA^3)	vdW (\AA^3)	V_i
Herringbone	Initial	350.50(2.76)	249.76(1.60)	1.40
	NN/GFN2-xTB	360.33(2.75)	251.17(1.67)	1.43
	NN/GFN-FF(rigid)	358.24(2.81)	250.91(1.71)	1.43
	NN/GFN-FF(flexible)	362.60(3.20)	250.80(1.71)	1.45
Parallel	Initial	350.85(3.66)	250.40(1.70)	1.40
	NN/GFN2-xTB	355.01(3.21)	251.12(1.57)	1.41
	NN/GFN-FF(rigid)	352.25(3.29)	250.96(1.65)	1.40
	NN/GFN-FF(flexible)	356.17(4.11)	251.12(1.69)	1.42

At the end of simulations, the pentacene monomers in the herringbone dimers show slightly larger increments of the Voronoi volume in the flexible crystal environments ($dV \approx 12 \text{\AA}^3$) than in the rigid crystal environments ($dV \approx 10 \text{\AA}^3$). The monomers in the parallel dimers show comparable changes in the Voronoi volumes in the flexible ($dV \approx 5 \text{\AA}^3$) and rigid ($dV \approx 4 \text{\AA}^3$) crystal environments. The vdW volumes in both dimers show small changes of less than 2\AA^3 . As a result, the V_i in both dimers is almost unchanged regardless of whether the crystal environment is fixed, suggesting that the pentacene crystal provides sufficient volume for the excited-state vibrations in the subpicosecond timescale.

We further compared the trajectories with rigid and flexible crystal environments. The $S_2 \rightarrow S_1$ decay time constants for the herringbone dimers are 61, 69, and 69 fs, obtained by the NN/GFN2-xTB, NN/GFN-FF, and NN/GFN-FF(flexible) models, respectively. We plot the trajectories below, where the black curves represent the averaged trajectories. The results of NN/GFN-FF(flexible) in panel (d) show the same structural changes as others in panels (b) and (c).

For the parallel dimers, the $S_2 \rightarrow S_1$ decay time constants are 33, 40, and 43 fs, respectively, obtained by the NN/GFN2-xTB, NN/GFN-FF, and NN/GFN-FF(flexible) models, respectively. We noted that the intermolecular distance R2 in the flexible crystal environment, shown in panel (d), is slightly longer than that in the rigid crystal environment, shown in panels (b) and (c). Both rigid and flexible crystal environment models show the same intermolecular stretching following R2, with a comparable period of 150 fs. The lateral displacements Rz are less affected than R2.

Overall, our supplementary results show that the ML photodynamics simulations with a rigid crystal environment reproduced the same nature of the excited-state vibrations of the pentacene dimer as those obtained from a flexible crystal environment. Thus, the rigid crystal environment is a reasonable approximation for studying the singlet fission dynamics of pentacene dimers. We added the above results in Supporting Information Section S9: Table S5 and Figure S19.

3) The lack of dynamical correlation in the casscf(4,4) calculations could affect the energetic order of the states, in particular favouring the quintet state with respect to the singlets. The authors should briefly comment on this.

- Answer: We computed the XMS6-CASPT2(4,4)/cc-pVDZ energies to quantify the role of dynamical correlation compared to the CASSCF potential energy curves. To reduce the limitation of the selected active space, we also computed the potential energies using the mixed-reference spin-flip time-dependent density functional theory (MRSF-TDDFT), J. Chem. Theory Comput. **2021**, *17*, 848–859, which can generate a spin-adapted multireference excited-state wavefunction including dynamical correlations. The plots below show the potential energy curves of the pentacene dimers with respect to intermolecular distances, as computed with SA6-CASSCF(4,4)/cc-pVDZ, XMS6-CASPT2(4,4)/cc-pVDZ, and MRSF-TDDFT/cc-pVDZ methods with the PBE0 functional. The choice of the PBE0 functional is based on its good accuracy in the recent benchmarks (DOI: 10.26434/chemrxiv-2024-k846p).

We note that the S₁ and S₂ states at both XMS6-CASPT2(4,4)/cc-pVDZ and MRSF-TDDFT(PBE0)/cc-pVDZ levels are significantly lower in energy than the SA6-CASSCF(4,4)/cc-pVDZ level due to the inclusion of the dynamical correlation. The energy of Q is higher than the S₁ and S₂ states. One possible reason for the positive Q-S₁ gap is that the Q states of the pentacene dimers are too localized to accommodate the Pauli repulsion. Therefore, the Q-S₁ gap must compute the delocalized Q states in a larger pentacene cluster or the pentacene crystal. Neither approach is feasible with current computational techniques. Thus, we removed our inaccurate discussions on the Q-S₁ gap to avoid misleading, which are *“The Q-S₁ gap becomes negative in the herringbone dimer when R1 > 5.1 Å (Figure 4a) and is negative in the parallel dimer at all R2 values (Figure 4b), suggesting the generation of an unbounded T-T biexciton at the S₁-FC region, followed by an exothermic triplet separation”*.

We added the above results to Supporting Information Section S2: Figure S3. In our answer to Reviewer 3's Question 1, we also discuss more details about the S₂-S₁ gaps in XMS6-CASPT2(4,4)/cc-pVDZ and MRSF-TDDFT/cc-pVDZ calculations.

- 4) In the simulated absorption spectra reported in figure S1, the band associated to the S₂ state is reported to be at lower energy with respect to the band associated to the S₁ state.

Why? My understanding is that it is implicitly assumed that the states labeled as S1, S2, S3, etc. are adiabatic singlet states in energetic order. Moreover, the bright state in figure S1 appears to be the lowest excited singlet state (in particular for the parallel dimer), which does not match with what is said in the main text. Am I missing something?

- Answer: Thank you for pointing out the unclear descriptions for the simulated absorption spectra. You are right about the state labels, which are the adiabatic singlet states. The plots below show the normalized distributions of the computed absorption wavelengths with the absorption bands for the herringbone and parallel dimer in panels (a) and (b).

The red and blue bands show the absorption intensities of S₁ and S₂ normalized to the maximum intensities of the S₂ band. The dashed lines indicate the distributions of the S₁ and S₂ wavelengths with equal oscillator strengths, which are normalized to the maximum value of the S₂ curves. In both dimers, the wavelengths of S₁ are longer than S₂ in the same geometry. Most low-lying S₁ wavelengths have zero intensities, while most S₂ wavelengths show significantly higher intensities. As such, the absorption spectra show the S₁ absorption band is higher than S₂, making S₂ the lowest accessible state by photoexcitation.

To understand the S₂ absorption and the small transition-allowed region in the S₁ absorption band, we plot the oscillator strengths as functions of the CSF weights, as shown below.

Our results show negative correlations between the TT characters and the oscillator strengths in both dimers. It confirms that the adiabatic S₁ and S₂ excited states become “dark” with a large TT character. The local excitation (LE) characters of each monomer display strong oscillator strengths in both S₁ and S₂, which are responsible for the

absorption band in the simulated spectra. Both herringbone and parallel dimers show stronger oscillator strengths in S_2 because of their greater LE weights than in S_1 . This results in larger intensities of S_2 than S_1 in the simulated spectra. The CT characters show no correlation to the oscillator strengths, but these configurations mix with the TT and LE. The $S_2 \rightarrow S_1$ decays can convert the LE characters to TT characters via mixing with the CT configurations. Therefore, our ML-photodynamics simulations from the S_2 -FC region inform the CT-mediated SF pathways.

On the other hand, the mixing of LE and TT mediated with CT also brings small transition-allowed regions in S_1 . These results agree with the proposed coherent SF pathway via direct excitation to the dark TT state (Ref 26 *Nat. Chem.* **2024**, *in press*, <https://doi.org/10.1038/s41557-024-01556-3>), which involved a charge resonance (CR) state (i.e., a symmetric superposition of the CT states) carrying finite transition probability borrowed from the main excitonic excitations. Note that the main excitonic excitation defined in this reference is equivalent to LE in our CSF representations. As such, our ML-photodynamics simulations from the transition-allowed S_1 -FC region elucidate the coherent SF pathways. Collectively, our results suggest the coexistence of CT-mediated and coherent SF pathways.

In addition, panels (a) and (b) show a larger mixing of CT and LE in the herringbone dimer than in the parallel dimer, shown in panels (c) and (d). This suggests the direct excitation to the TT state could also be anisotropic according to the direction of the pentacene dimers.

We added the above results to the Supporting Information in Section S4: Figure S5 and Figure S6. We also included brief discussions on the SF mechanisms via the CT-mediated and coherent pathway in the main text, which reads, “The simulated absorption bands (Figures 2c and 2d) show zero intensities at most low-lying wavelengths of the adiabatic S_1 state (Figure S5), due to the dominant TT character (Figure S6). They result in the lowest optical bright states being the adiabatic S_2 state with a local excitation (LE) character. The mixing of LE and TT with CT configurations in the S_2 -Franck-Condon (FC) points suggests that S_2 could turn into a TT state via $S_2 \rightarrow S_1$ transitions (Figure S6). Thus, the photoexcitation to the adiabatic S_2 state informs the CT-mediated SF pathways. The adiabatic S_1 state displays minor transition-allowed regions with the help of the CT-mediated mixing of LE and TT characters (Figure S6). The photoexcitation in this region could directly generate the TT state, corresponding to the coherent SF pathways. The above results suggest a coexistence of the CT-mediated²⁵ and coherent²⁶ SF pathways reported in recent experiments. Therefore, we perform ML-photodynamics simulations⁴⁵ to study the SF mechanisms in both pathways.” on Page 4.

5) A possible typo: The acronym CASSCAF, appearing several times in the manuscript and in the SI, should be replaced by CASSCF? If not, please specify.

- Answer: Thank you for finding this typo. We have corrected it accordingly.

Reviewer #2 (Remarks to the Author):

This manuscript reports a computational study of singlet fission in pentacene crystals. The authors applied a multiscale multiconfigurational approach, combined with machine learning, to simulate the dynamics of singlet fission in pentacene crystals. This is a nice, albeit incremental, improvement in our understanding of singlet fission in this most widely studied model system. The claim that "singlet fission mechanism in pentacene crystals is unresolved" is overstated. Extensive work in the last 15 years, both experimental and computational, in pentacene crystals and model dimers, has established perhaps the clearest mechanistic picture on singlet fission in this model system. The role of vibrational motions, particularly intermolecular motions, has been extensively studied. Moreover, the sub 100 fs dynamics determined in experiments have also been reproduced in many computational/simulation works, including a number of reports based on multiconfigurational interactions.

I would agree that the current work may have computational advantages in adding to this mechanistic picture, but it is by no means the final say. The model adopted by the authors, i.e., a pentacene dimer embedded in a rigid matrix, is also an approximation. Exciton delocalization in pentacene crystals is likely more extensive than what is captured by the dimer (see, e.g., J. Phys. Chem. Lett. 2013, 4, 2197). Such delocalization is reflected in the proposals of CT states in singlet fission, culminating in the recent experimental verification (Nature 2023, 616, 275, ref. 25). The sampling of some low frequency modes, particularly the intermolecular motion, in assisting conical intersection type of dynamics, has its own limitations. The actual involvement of vibrational motions may be highly multidimensional. In this sense, the claim of novelty and importance is a stretch. I am not an expert on machine learning, but the author's approach likely improves computational efficiency. Report of such improvements belongs to a technical journal, not the broader readership of Nat. Comm.

- Answer: Thanks for your comments. When we reviewed our manuscript from this viewpoint, we realized that many of the concerns may stem from confusing points that led to misunderstanding. As such, we have worked hard to provide additional information and reworked some text to address the raised concerns. We believe that our work is novel because 1) our calculations confirmed the coexistence of the CT-mediated and coherent SF pathways in the pentacene crystal, which have been debated in recent experimental reports; 2) our simulations revealed competing SF channels in the herringbone and parallel dimers, explaining the anisotropic SF phenomena in pentacene crystal; 3) we provided a comprehensive analysis on the relationship between the electronic configurations and intermolecular motions, which identified a quasi-one-dimensional intermolecular motion that elongates the intermolecular distances of the herringbone and parallel dimers to form the TT state in both CT-mediated and coherent pathways. Together, this work delivers a unified and simplified SF mechanism for pentacene systems.

Comments 1: *This is a nice, albeit incremental, improvement in our understanding of singlet fission in this most widely studied model system.*

- Answer: We agree that this work improved our understanding of SF in the pentacene crystal, a widely studied system. The advances in this work go far beyond reemphasizing the existing mechanistic knowledge or reporting an efficient simulation technique. This manuscript provides a unified perspective of the SF mechanisms that reduces the ambiguity in matching the experimental observations and theoretical models. According to Ref 23-26, the experimental techniques enabled us to observe SF at the orbital-level resolution (Ref 25). A more recent work (Ref 26) showed an alternative SF pathway via direct excitation to the dark TT state. However, due to a lack of structural evidence, these results still could not fully answer whether and how molecular vibrations control the different SF pathways. The relationships between the molecular vibrations and the anisotropic SF phenomena are unclear. On the other hand, recent theoretical studies employed the molecular or crystal models, which either have no dynamical information (Ref 33-38) or cannot describe the multiconfigurational nature of the excited state, failing to explain the origin of the TT state (Ref 41), or overestimate the SF time constants, resulting in inaccurate vibration information (Ref 42-44).

The methodology we apply here is groundbreaking for this problem; it uses the multiconfigurational method that accurately quantifies the CT, LE, and TT characters of the excited states associated with the molecular vibrations directly sampled from dynamics simulations for the first time. The ML photodynamics simulations reproduced the experimentally reported SF time constants, which provided high-fidelity electronic and atomistic information to resolve the role of the molecular vibrations in SF. The different SF dynamics discovered in the herringbone and parallel dimer models provide a straightforward explanation for the anisotropic behavior, which is otherwise unanswered in the literature. Our simulated absorption spectra using multiconfigurational calculations confirmed the coexistence of the CT-mediated and coherent SF pathway. Thus, our work not only leverages state-of-the-art computations with ML acceleration but also bridges the gap between all experimental and theoretical findings reported to date; we are unaware of any other SF study approaching this level of novelty.

Comments 2: *The claim that "singlet fission mechanism in pentacene crystals is unresolved" is over stated.*

- Answer: Thank you for pointing out our imprecise claim. We agree that our previous claim about the current situation of the SF study is overstated. In literature, Ref 25 (*Nature* **2023**, 616, 275–279), states,

"Intense efforts have been invested in explaining the primary step, the formation of ¹TT. However, even in the most studied and highly efficient SF system, pentacene, ambiguity prevails over its mechanism...Berkelbach et al. proposed a mechanism based on delocalized charge-transfer (CT) states...In contrast to the CT-mediated mechanism, Chan et al. postulated a coherent mechanism...Furthermore, an explanation based on a conical intersection mechanism has emerged."

Even with the challenging orbital-resolved spectroscopic results, Ref 25 concluded,

“Our observations show the nature of the electronic states and reveal the significant CT character of the singlet exciton...Our observations do not exclude a vibronic or a conical intersection mechanism based on high-frequency modes.”

A more recent Ref 26 highlights the ongoing debates on the SF mechanism,

“The formation of $1TT$ through SF can be extremely rapid (<100 fs), despite the state being optically dark and generally unable to couple strongly to the bright singlet S_1 , and the detailed mechanism of this process is actively debated.”

However, Ref 26 provided a different viewpoint from Ref 25,

“we revive the concept of a coherent SF pathway...We demonstrate direct photoexcitation of dark $1TT$ as a general phenomenon across a range of pentacene derivatives, indicating that photoexcitation of bright S_1 is no longer required to access the entangled pair.”

Given the above competitive viewpoints in recent studies, we find the SF mechanism of pentacene has still been disputed. Thus, we change the word “unresolved” to “disputed” in our Abstract. Moreover, our ML-photodynamics simulations shows the “contrasting” CT-mediated or coherent SF mechanisms co-exist in the pentacene crystal.

Comments 3: *The role of vibrational motions, particularly inter-molecular motions, has been extensively studied.*

- Answer: We agree that the literature includes investigations of vibrational motions. However, a quantitative relationship between the intermolecular motions and the SF mechanisms has not been established until this manuscript.

Ref 24 (*J. Phys. Chem. Lett.* **2021**, *12*, 3142–3150) identified a vibration mode of the pentacene tetramer responsible for the anisotropic SF phenomena. The vibrational mode is too delocalized to determine the role of each monomer, leading to the question about the relationship between the excited-state characters and the specific molecular motions raised by Reviewer 3.

Ref 36 (*Nat. Chem.* **2010**, *2*, 648-652), Ref 38 (*J. Am. Chem. Soc.* **2014**, *136*, 5755–5764), and Ref 47 (*J. Am. Chem. Soc.* **2011**, *133*, 19944–19952) scanned the excited-state electronic configurations along with the intermolecular distance of the herringbone dimer. However, the investigated motions required further justification by the excited-state dynamics, which motivated us to explore the dynamics.

Ref 59 (*Sci. Adv.* **2020**, *6*, eabb0052) reported the wave-packet dynamics of the pentacene dimer; however, their analysis was limited to a few vibrational modes (i.e., intermolecular motions), thus overestimating the SF time scale.

Our ML photodynamics simulations generated comprehensive structural information on the molecular motions that participated in the SF of pentacene dimers. The agreement with the experimental time scale enables us to quantify the excited-state characters associated with the molecular motions that facilitate the SF process, showing a quantitative picture of the SF mechanism.

Comments 4: *The sub 100 fs dynamics determined in experiments have also reproduced in many computational/simulation work.*

- Answer: We agree with that, nowadays, computational works produce SF timescales comparable to experiments. We believe the SF timescale agreement is necessary but insufficient to understand the SF mechanisms.

Recent literature, including Ref 59(*Sci. Adv.* 2020; 6: eabb0052), Ref 44 (*J Chem. Phys.* 2023, 159, 224301), and *J. Phys. B: At. Mol. Opt. Phys.* 2024, 57, 105101, reported the SF dynamics around 100 fs. However, their dynamics are limited to the selected vibrational modes or a model Hamiltonian, where the specific role of the low-frequency molecular motions in SF is not fully explained. None of the dynamics simulations reported in the literature explain the origin of the anisotropic SF phenomena at the multiconfigurational quality in full dimensionality.

Our manuscript includes the first full-dimensional dynamics of pentacene dimers at the multiconfigurational quality. Our trajectory analysis further decomposes the vibrational modes to chemically intuitive molecular motions (please refer to our answer to Reviewer 3's Question 8 for details) to determine the structure-property relationship between the specific molecular motions and the excited-state characters in dynamics, which provides a deeper understanding of the SF mechanism.

Comments 5: *Exciton delocalization in pentacene crystals is likely more extensive than what is captured by the dimer (see, e.g., J. Phys. Chem. Lett. 2013, 4, 2197).*

- Answer: Thank you for reminding us of this important reference. The references reported an average electron-hole distance for the singlet excited states of 6–8Å. The figure below shows the intermolecular distances between the monomers. The dimer model can describe the CT reported at 6Å, while computing the CT at 8Å requires a tetramer model.

This reference also reported the average electron-hole distances of 2\AA in the triplet state. It suggests the SF will rapidly collapse the delocalized singlet exciton to a localized triplet pair state, reported in Ref 59 (*Sci. Adv.* **2020**, 6: eabb0052). Ref S3(*Phys. Rev. Lett.* **2017**, 119, 267401) shows clear singlet exciton wavefunctions, where the electrons mainly delocalized over the monomers neighboring to the hole on the central pentacene. Thus, the dimer model could capture most CT characters of the singlet exciton. Given the excellent agreement in the SF time constants between the reference (30-70fs) and our works (33-61fs), we conclude that the dimer model reasonably approximates the SF processes of pentacene crystal.

We included the above clarifications in Supporting Information Section S1.

Comments 6: *Such delocalization is reflected in the proposals of CT states in singlet fission, culminating the recent experimental verification (Nature 2023, 616, 275, ref. 25).*

- Answer: Thank you for highlighting the importance of this experimental work. As our answer to your Comment 5, the dimer model is a cost-effective approximation for studying the CT and TT characters in SF dynamics. Our multiconfigurational calculations based on the dimer model confirmed the large CT characters in the lowest optical bright state (i.e., the adiabatic S_2 in our calculations), which is in line with the above reference. Moreover, our calculations show a partial mix of CT and TT characters in the dark state (i.e., the adiabatic S_1) due to the intermolecular vibrations, which allows a direct electronic transition from the ground state to the dark TT state for SF. These findings agree with the most recent experiment that observed the direct excitation to the TT state (*Nat. Chem.* 2024, in press, <https://doi.org/10.1038/s41557-024-01556-3>). Please refer to our answers to Reviewer 1's Question 4 for details.

Comments 7: *The sampling of some low frequency modes, particularly the inter-molecular motion, in assisting conical interception type of dynamics, has its own limitations.*

- Answer: We agree that our approach did not achieve 100% agreement between computations and experiments; we are not sure that is possible given the complexity of the problem we are solving! We aim to minimize the gap to the best of our efforts. Instead of starting from a pure model, we collected the structures from the dynamics simulations, where the results showed agreement with the experimentally observed SF time constants. We sampled the low-frequency vibrational modes by evaluating

their contributions to the simulated trajectories to reduce human bias in the mode selection. Then, we designed the chemically intuitive motions informed by the important vibrational modes. Finally, we discovered the relationship between the CT and TT characters and the intermolecular motions. Our findings explained the role of CT and TT states in the SF processes and the origin of the anisotropic phenomena. We use these results to survey and bring a deeper understanding to those reported experimental and computational results.

Comments 8: *The actual involvement of vibrational motions may be highly multidimensional.*

- Answer: Thank you for suggesting an interesting and possible role of the molecular vibrations in the SF mechanisms. However, our trajectory analysis revealed that the formation of the TT state in the SF of pentacene crystal is mainly driven by quasi-one-dimensional motions elongating the intermolecular distances in both herringbone and parallel dimers. The nearly orthogonal orientations of the two dimers produce the anisotropic SF phenomena. Please refer to our answers to Reviewer 3's Question 8 for details.

Reviewer #3 (Remarks to the Author):

The work "Machine learning photodynamics beyond the Frenkel exciton model: Intermolecular vibrations trigger ultrafast singlet fission in pentacene crystal" by Li et. al. describes a novel and interesting way to simulate the photodynamics of a pentacene crystal using trajectory surface hopping without the use of the Frenkel exciton model. Instead a multiscale cluster approach was used in combination with multiconfigurational CASSCF calculations of the relevant pentacene dimers (herringbone and parallel dimer). To speed up the computations and to enable trajectory surface hopping simulations in a reasonable time frame, neural networks were employed to replace the (expensive) CASSCF calculations. The resulting decay rates are in excellent agreement with the experimental values for both the herringbone and the parallel dimers. The manuscript is generally well written and but the reviewer suggests its publication in Nature Communications only after the following questions/comments are addressed.

1. The authors write "This method was previously benchmarked against multireference methods with second-order perturbative corrections, which showed essentially the same topology in the excited-state potential energy surfaces (PES) for the pentacene dimer." Here they cite Ref. 45 "Mechanism for Singlet Fission in Pentacene and Tetracene: From Single Exciton to Two Triplets" from Zimmerman et. al. However, inside this paper, I could not find calculations with a second-order perturbation theory method on the pentacene dimer, neither using CASPT2 nor multi-reference perturbation theory (MRMP). Instead the computational efficient restricted active space double spin-flip (RAS-2SF) method was compared with CASSCF (see SI of Ref. 45) and both shifted to the MRMP(12,12) values of the pentacene monomers at the dissociation limit. The authors of Ref. 45 write, that they compare with CASSCF but do not further specify the computational method. In a cited former publication of some of the same authors (<https://doi.org/10.1038/nchem.694>) "Singlet fission in pentacene through multi-exciton quantum states" it is stated that for the PE scans CASSCF(8,8) was used

for the dimer. Therefore, I assume this was also used for the scan in the SI of Ref. 45. Where was the benchmark with respect to the multireference methods with second-order perturbation corrections done, as I could not find it in Ref. 45?

- Answer: Thank you for carefully reviewing the computational details in the cited literature. We agree that the literature did not report direct comparisons between the CASSCF and multireference methods for the potential energy curve. Instead, they shifted the CASSCF energy to the MRMP(12,12) values at the dissociation limit. As it is essential to learn the dynamical correlation in the excited-state potential energy surface, we perform the XMS6-CASPT2(4,4)/cc-pVDZ and MRSF-TDDFT(PBE0)/cc-pVDZ calculations to benchmark the CASSCF results, shown in the plots below.

The main text mentions that the ω B97XD/def2-TZVP calculations optimized the S_0 minima of herringbone and parallel dimers at $R1 = 5.84 \text{ \AA}$ and $R2 = 3.70 \text{ \AA}$. The XMS6-CASPT2(4,4)/cc-pVDZ calculations show underestimated distances at $R1 = 5.6 \text{ \AA}$ and $R2 = 3.5 \text{ \AA}$. Although the dynamical correlation lowers the excited-state energies, the values (herringbone: 2.48 eV; parallel: 2.63 eV) are still higher than the experimental values of the pentacene crystal (1.83 eV) in the abovementioned reference (our Ref 36). These inconsistent results suggest that the XMS6-CASPT2(4,4)/cc-pVDZ method cannot capture all essential dynamical correlations, possibly due to the limited active space. The MRSF-TDDFT(PBE0)/cc-pVDZ calculations show the S_0 minima at $R1 = 6.1 \text{ \AA}$ and $R2 = 3.9 \text{ \AA}$. The resulting vertical excitations are 1.79 eV and 1.89 eV for the herringbone and parallel dimers, close to the experimental value. As such, we chose the MRSF-TDDFT(PBE0)/cc-pVDZ results as the reference in our benchmarks. The SA6-CASSCF(4,4)/cc-pVDZ potential energy curves show close-lying S_1 and S_2 states and the S_0 minima at $R1 = 6.1 \text{ \AA}$ and $R2 = 4.0 \text{ \AA}$, which are consistent with the MRSF-TDDFT(PBE0)/cc-pVDZ results. Thus, the SA6-CASSCF(4,4)/cc-pVDZ calculations are sufficient for studying the $S_2 \rightarrow S_1$ decay.

We added the above information to Supporting Information Section S2: Figure S3.

Was the choice of CASSCF(4,4) based on the RAS-(4,4)-2SF method in Ref. 45? If so, how do the two methods compare? And how well is it compared to the CASSCF(8,8) used in Ref. 20 of Ref.45?

- Answer: We choose the (4,4) active space according to chemical intuition. The electronic excitation of the pentacene monomer mainly involves the electronic transition from the highest occupied molecular orbital (HOMO) to the lowest unoccupied molecular orbital (LUMO), giving a (2,2) space. The minimal active space of the pentacene dimer is (4,4). We confirmed the selected orbitals belong to the HOMO and LUMO of both monomers by performing the orbital localization (Figure 1c).

We compare the (8,8) and (4,4) space by computing the potential energy curve following the elongation of the intermolecular distances, shown below.

The SA6-CASSCF(8,8)/cc-pVDZ calculations show notably larger S_2 - S_1 gaps than that with the (4,4) space, which overestimates the S_2 - S_1 gaps compared to the MRSF-TDDFT(PBE0)/cc-pVDZ reference. Thus, the SA6-CASSCF(8,8)/cc-pVDZ results must be corrected by including the dynamical corrections. The (4,4) space results agree with the reference results better than the (8,8) space, encouraging us to use it for the NN training data calculations and the ML photodynamics simulations.

We added the above information to Supporting Information Section S2: Figure S3.

As the authors apply a machine learning (ML) approach to generate the CASSCF energies, why did they not use a higher level of theory like CASPT2?

- Answer: The main reason we could not use CASPT2 data of pentacene dimers is the infeasible computational costs. Our previous work (*Chem. Eur. J.* 2022, 28, e202200651) trained NNs with XMS-CASPT2 data for small molecules with only 12 atoms. Another work by Westermayr et al. (*Nat. Chem.* 2022, 14, 914–919) trained NN with CASPT2 data from tyrosine, which only has 24 atoms. The pentacene dimers contain 72 atoms, substantially increasing the computational cost of the CASPT2 calculations, especially for the gradient calculations.

Along these lines, how much more expensive would be the CASPT2 calculations?

- Answer: We performed the SA6-CASSCF(4,4)/cc-pVDZ calculations using BAGEL, which were efficiently parallelized with 4 CPUs. The XMS6-CASPT2(4,4)/cc-pVDZ calculations with BAGEL failed because they require more than 500 GB RAM for a single calculation. Thus, we switched to OpenMolcas for the XMS6-CASPT2(4,4)/cc-pVDZ calculations, accelerated with 20 CPUs. The MRSF-TDDFT(PBE0)/cc-pVDZ calculations used the underdeveloped code, OpenQP[DOI: 10.26434/chemrxiv-2024-k846p], paralleled with 4 CPUs. The table below lists the averaged computational costs for the single-point calculations.

Method	Time (s)	with gradient	CPUs
SA6-CASSCF(4,4)/cc-pVDZ	5205	no	4
SA6-CASSCF(4,4)/cc-pVDZ	9032	yes	4
XMS6-CASPT2(4,4)/cc-pVDZ	92032	no	20
MRSF-TDDFT(PBE0)/cc-pVDZ	2295	no	4
MRSF-TDDFT(PBE0)/cc-pVDZ	5090	yes	4

Computing the energy and gradient of one state at the SA6-CASPT2(4,4)/cc-pVDZ level costs 1.7 times higher than the energy-only calculation. The XMS6-CASPT2(4,4)/cc-pVDZ energy calculation requires more than 1 day, which is about 18 times longer than the SA6-CASSCF(4,4)/cc-pVDZ calculation. The gradient calculations at the XMS6-CASPT2(4,4)/cc-pVDZ exceed our affordable computational resources. The MRSF-TDDFT(PBE0)/cc-pVDZ calculations show promising efficiency, where the energy calculation without and with gradients only took 44% and 56% of the time spent by the SA6-CASSCF(4,4)/cc-pVDZ calculations. The efficiency of the MRSF-TDDFT(PBE0)/cc-pVDZ calculations is realized by the newly designed quantum chemistry code, OpenQP (DOI: 10.26434/chemrxiv-2024-k846p), which is still under development and is only available for benchmark purposes during this revision. Nevertheless, the SA6-CASSCF(4,4)/cc-pVDZ shows a good balance between accuracy and efficiency.

We added the above timing information in Supporting Information Section S2: Table S1.

In Ref. 20 of Ref. 45 it is stated that due to the lack of dynamic correlation in CASSCF an overestimation of the energy of the S₁ occurs and it is needed to shift the energy of the electronic states accordingly to the MRMP(12,12) results. Do the authors see a similar behavior in their calculation?

- Answer: Indeed, the lack of dynamical correlation overestimates the S₁ energy at the SA6-CASSCF(4,4)/cc-pVDZ level. We also want to point out that the singlet fission of

the pentacene dimer mainly involves the $S_2 \rightarrow S_1$ decay. The ML photodynamics simulations require accurate S_2-S_1 gaps rather than accurate magnitudes of the S_1 energies, as long as the curvature of the S_1 surface is correct. According to our benchmarks, the SA6-CASSCF(4,4)/cc-pVDZ method reproduces a similar topology of the S_1 and S_2 potential energy curves to the multireference methods, especially for the close-lying S_2-S_1 gaps. Therefore, the S_1 and S_2 energies need no shift to multireference results.

As the hopping rate is strongly effected by the energy gap between the relevant electronic states, are all excited states simply shifted by a constant factor between the CASSCF and the multireference methods with second-order perturbative corrections, or are there changes in the energy gap between the two methods?

- Answer: We do not shift the energies at the SA6-CASSCF(4,4)/cc-pVDZ level. Our benchmarks against multireference methods show that the SA6-CASSCF(4,4)/cc-pVDZ method reproduces a similar topology of the S_1 and S_2 potential energy curves to the multireference results, especially for the close-lying S_2-S_1 gaps. Thus, the accuracy of SA6-CASSCF(4,4)/cc-pVDZ level suffices the requirements of ML photodynamics simulations for the singlet fission of the pentacene dimers.

2. One difficulty in learning QM/MM data is to learn the influence of the MM sites on the QM structure, as there are many more configurations to consider. Are the MM (xTB) atoms included in the NN fit or is a pure ONIOM approach used, where only the QM atoms are included?

- Answer: We do not include the xTB atoms directly in the NN training. Instead, we include their influence as background charges. Our answer to Reviewer 1's Question 2 shows that the rigid crystal environment is suitable for simulating the excited-state dynamics of the pentacene dimers. Thus, we embedded constant background charges in all training data calculations. This allows the NN to implicitly learn the energies and gradients affected by the charges of the xTB atoms and does not need to include additional configurations for the charge positions. The background charges are obtained using the restrained electrostatic potential (RESP) method at the ω b97xd/def2-TZVP level.

We compute the total energy using the electrostatic embedding ONIOM approach (*J. Chem. Theory Comput.* 2019, 15, 4, 2504–2516). The NN-predicted energies already include the electrostatic interactions with the xTB atoms. The subtracted GFN2-xTB energies of the pentacene dimers also include the same background charges, so the electrostatic interactions will not be double-counted.

We clarified our description as “In training data and the PES scan calculations, the pentacene dimers were computed with the SA6-CASSCF(4,4)/cc-pVDZ calculations using the BAGEL program⁶⁷ with the same RESP charges, where only the pentacene dimers are included to train NN” in Computational Methods on Page 12.

3. The authors state, that a single ML surface hopping dynamics was completed in 5 days due to the computational bottleneck of the GFN2-xTB calculations. Is the use of GFN2-xTB necessary? Especially, when the ML approach is used or did the authors consider more approximate but computational cheaper schemes like the GFN-FF approach?

- Answer: Thank you for the excellent suggestion on testing the GFN-FF approach. We performed NN/GFN-FF photodynamics simulations for both dimers. The NN/GFN2-xTB calculations spent 991.8s for one step using 6 CPUs, while the NN/GFN-FF calculations finished in 4.7s with 6 CPUs, which provides an additional 213-fold acceleration. The resulting acceleration to the SA6-CASSCF(4,4)/GFN2-xTB calculations becomes 5099-fold. As we showed in our answers to Reviewer 1's Question 2, the GFN-FF approach produced similar results to the GFN2-xTB approach. The main difference is that the GFN-FF calculations gave slightly longer time constants (herringbone: 69 fs; parallel: 40) than the GFN2-xTB calculations (herringbone: 61 fs; parallel: 33). Our results suggest that the GFN-FF method could be a good candidate for ONIOM calculations.

We add a note for this finding in Computational Methods on Page 13, which reads “We found the GFN2-xTB and GFN-FF⁷⁰ methods produced similar results in the ML-photodynamics simulations (Figure S19). A single ML-photodynamics trajectory computed with the ee-ONIOM(SA6-CASSCF(4,4)/cc-pVDZ:GFN2-xTB) requires 110 days using 6 CPUs. The ee-ONIOM(NN/GFN2-xTB) and ee-ONIOM(NN/GFN-FF) calculations finished in 4.6 days and 0.5 hours, corresponding to a 24-fold and 5099-fold acceleration”.

4. For the state analysis and to better characterize the final state of the trajectory, did the authors recompute (a selection of) the final structures obtained with the ML learning surface hopping approach and perform ee-ONIOM(SA6-CASSCF(4,4) or CASCI/cc-pVDZ:GFN2-xTB) calculations on them? Or was the analysis purely done based on structural information (e.g. the R1/R2 values)? If only structural information was used, it would be good to verify at least for a selection of the final structures of the dynamics that the final state (S1) is indeed a triplet state. This point is crucial, as the singlet fission rate is defined as the S2->S1 decay rate.

- Answer: This is a very important point. We recomputed the electronic structures of the final snapshots with ee-ONIOM(SA6-CASCI/cc-pVDZ:GFN2-xTB) for all trajectories. The table below lists the average and standard deviation of the weights for the TT states in the final snapshots.

Dimer	Initial state	average TT weights	standard deviation
Herringbone	S ₂	0.67	0.0028
	S ₁	0.67	0.0009
Parallel	S ₂	0.67	0.0010
	S ₁	0.67	0.0010

Our results show that the TT state dominated the S₁ state at the end of the simulations from the S₂-FC points and the transition-allowed S₁-FC points. The average weight of 0.67 reached the maximum value we observed in the intermolecular distance scan (Figures 4e and 4f). The minor standard deviations imply all trajectories arrived at the TT states. Therefore, the S₂→S₁ decay time constants reflect the CT-mediated singlet fission time constants and the S₁ dynamics corresponds to the coherent singlet fission.

We added the above results in Supporting Information Section S5: Table S4.

5. The authors write that the average value of R1 is 5.92 Angstrom at the S₂/S₁ surface hopping points in herringbone dimers. What are these "surface hopping points"? Are these structures at which a hop occurs?

- Answer: Yes. The surface hopping points refer to the structures where the surface hopping occurs. We added a clarification to this phrase on Page 6, which reads "5.92 Å at the S₂/S₁ surface hopping points (*i.e.*, the structures where S₂→S₁ transitions occur)".

6. The authors write that the ML-photodynamics simulations "are transferable to understanding the SF process in the crystals of pentacene and hexacene." However, in the manuscript only pentacene results are shown. Is the statement regarding hexacene a (valid) guess or did the authors perform hexacene calculations as well?

- Answer: Thank you for commenting on our hypothesis regarding the SF mechanism of hexacene. We did not mean to conclude our simulations are "transferable" to understanding the SF process of hexacene. We wanted to highlight the similarity of the anisotropic SF phenomena of hexacene and pentacene crystals, possibly undergoing similar mechanisms as we observed in the ML photodynamics simulations for pentacene dimers. We did not perform ML photodynamics simulations for hexacene crystals. Since our hypothesis was not verified, we found our previous argument is premature. We removed "~~Thus, the computationally elucidated SF mechanisms found in our ML-photodynamics simulations are transferable to understanding the SF process in the crystals of pentacene and hexacene~~" on Page 6. We clarified our hypothesis, "Previous experiments also reported an anisotropic SF in the hexacene crystal with a factor of 4,²¹ where we anticipate a similar anisotropic SF mechanism to the pentacene" on Page 6.

7. For the neural network calculations, both energies and forces are predicted based on QM data. One issue, when energies and forces are both fitted by the neural network is that the gradients are not the analytic derivatives of the energies, which can lead to issues during the dynamics (e.g. non conserved total energies). Did the authors encounter this problem? One way to prevent this is to only fit the energies using the NN and compute the gradient analytically from the NN using an automated gradient approach (e.g. autograd of PyTORCH). Did the authors also try this approach?

- Answer: You are correct that training NN potential requires fitting energy and forces together and using the analytical gradient of NN to fit the forces. Our NN models are implemented using exactly this approach. We added this clarification in the Computational Methods, "The NN computes the inverse distance matrix of the input molecule to predict the energies and gradients, where the atomic gradients are obtained from the analytical gradients of the NN" on Page 12 and Supporting Information.

8. How exactly are the distances R1 and R2 defined? The authors write: "The intermolecular distances are defined by the carbon atoms in each central ring." I assume, the definition is the one in Figure S3 of the supplementary material, but from the text and Figure 1, it would not be clear to me. It might be helpful to point to Fig. S3 a)/d) just for clarification.

- Answer: Thank you for addressing the unclear definitions of the R1 and R2 distances. We revised Figure 1b to show the definitions of the R1 and R2, as shown below.

This definition of the distances raises the question, how the displacements were done for the scans in Figure 4. I assume, the pentacene molecules were displaced along the R1/R2 vector, while keeping all the other coordinates frozen?

- Answer: You are correct. We performed rigid potential energy scans along with the R1 and R2 vectors in Figure 4 (now Figure 5). We did not optimize the structures of dimers because constraining the R1/R2 distances removes most intermolecular degrees of freedom, while the remaining intramolecular motions are less relevant to the SF

processes according to our dynamics results. To obtain a complete picture of the excited-state configurations as a function of dimer structures, we perform the same potential energy scan and electronic configuration analysis for the rotations and lateral shifts of the monomers, as suggested by your following questions. Please refer to our answers to your following questions for details.

This in fact represents a translation between the two pentacene molecules. The authors show nicely, how an increase in R1/R2 distance change the state character of the S1 to a triplet state, however, during the dynamics (Figure 2), how much of the changes of R1/R2 are due to this translation and how much is due to other coordinate changes e.g. rotations especially along the Z-Axis (as defined in Figure 3 c/d) through the center of mass of the individual pentacene molecule?

- Answer: Thank you for pointing out the importance of quantifying the contributions of intermolecular motions in the excited-state dynamics. In Figures 4c and 4d, the dominant vibrational modes in trajectories show the elongation of the intermolecular distances. However, the vibrational modes are also mixed with other motions, which makes it difficult to estimate the contributions of the intermolecular translations. To quantify the contributions from different intermolecular motions to the excited-state vibrations, we define four chemically intuitive motions corresponding to three orthogonal translations (M_x , M_y , M_z) and one rotation (M_r) of the monomers, where M_x follows the directions of the R1/R2 vectors, as shown below.

We first project the vibrational modes that dominate the trajectories (Figures 4c and 4d) to the defined motions. The projection norm represents the contributions of each motion.

The panels (a) and (b) show substantial contributions from M_x and M_z to the 70 cm^{-1} and 45 cm^{-1} modes of the herringbone dimers, respectively. The panels (c) and (d)

show M_x and M_z also dominate the 50 cm^{-1} and 29 cm^{-1} modes of the parallel dimers. These results suggest the elongation of intermolecular distance (M_x) and the lateral motions (M_z) are essential to understanding the $S_2 \rightarrow S_1$ decay mechanism. Figures 3c and 3d in the main text indicate the participation of the other vibrational modes in the excited-state dynamics. To determine the total contributions of the defined motions in the dynamics, we computed the structural changes of the dimers at the end of the simulations from the initial conditions. We plotted their projection norm in the figures below.

Our results show the trajectories of both herringbone and parallel dimers are governed by the elongation of the intermolecular distances (M_x). The contributions of M_y and M_z are comparable, while the rotations are minimal. Our discussions in Figure 5 revealed that the S_1 configurations change from CT state to TT state with increasing distance along with M_x (i.e., R1/R2). However, our previous version did not investigate the relationships between the S_1 characters and the rest of the motions (e.g., M_y , M_z , and M_r), which makes our SF mechanism incomplete. Therefore, we performed the rigid potential energy scans and electronic configuration analysis for the rest of the motions in the herringbone and parallel dimers.

The figures in the first row illustrate the potential energy curves of the herringbone dimers displaced by (a) M_y , (b) M_z , and (c)-(d) M_r of individual monomers. The S_0 minima structures are at the middle point of the scans. The S_2 and S_1 states are closely and show increasing energies following positive and negative displacements. These results suggest these motions are not responsible for the energy relaxations in S_2 and S_1 states. The potential energy curves following M_z are relatively flat, which

explains the lateral motions observed in the dynamics. The figures in the second row show the number of unpaired electrons (N_e) and the weights of the TT character in the S_1 state following the scan. M_y and M_z do not notably affect the unpaired electrons or the TT characters. The rotation of monomer 1 slightly reduced N_e and the TT character in the S_1 state when it is perpendicular (i.e., rotation angle > 0) to the other in panel (g); the rotation of monomer 2 also decreased the TT characters in S_1 when the C-H bonds are rotating toward (i.e., rotation angle > 0) the benzene rings in panel (h). Nevertheless, the TT characters are still more than 50%, dominating the nature of the S_1 state.

The above figures plot the potential energy curves, N_e , and TT characters of the parallel dimers. The S_2 and S_1 energies show similar features as the herringbone dimers. The almost flat potential energy curves in M_z directions explain the lateral motions observed in the dynamics. The N_e and TT characters remained nearly unchanged in all scans. Together with the results of the herringbone dimers, our calculations suggest M_y , M_z , and M_r could not significantly affect the TT characters of the S_1 state. Therefore, the elongations of the intermolecular distances (M_x) are the driving force that generates the TT state in the pentacene dimers.

We added the above results in Supporting Information Sections S8: Figure S13–S17.

In fact in Figure 3, the authors identify two low lying intermolecular rotations as the main vibronic active low-frequency motions. Could the authors comment on the change in excited state character of the system along these rotation?

- Answer: Thank you for raising this important question. The low-frequency vibrations have both translation and rotation components. To better quantify their contribution to the vibrations and their roles in changing the excited-state character, we introduce four chemically intuitive motions corresponding to three orthogonal translations (M_x , M_y , M_z) and one rotation (M_r) of the monomers. We evaluated the contributions from these motions to the vibrational modes that dominate the excited-state dynamics of pentacene dimers and their contributions to the structural changes throughout the entire trajectories. We further explore the excited-state potential energies and electronic configurations in the positive and negative displacements following these motions. Our results show that only the motions elongating the intermolecular distances R1 and R2 could effectively enhance the TT characters in the S_1 state

(Figures 5e and 5f). In contrast, other motions barely affected the character of the S_1 state. Please refer to our answers to your previous questions for a comprehensive discussion of the related results.

Minor remarks:

page 4: I assume, that the SA6-CASSCAF(4,4) is a typo and should be SA6-CASSCF(4,4), if not please explain the difference between these methods. If yes, please change this also in the other cases.

- Answer: Thank you for addressing this typo. We have corrected it accordingly.

page 8: "Since the trajectories showed efficient S2/S1 surface hoppings to continue the SF process". Better use something like: "population transfer from S2 to S1" instead of "S2/S1 surface hoppings", if these is meant.

- Answer: Thank you for the excellent suggestion. We have removed the above phrase in our revised discussions on Page 6.

The link in the SI (https://github.com/mlcclab/PyRAI2MD_publications/Pentacene) does not work and should be updated.

- Answer: Thank you for carefully reviewing our code and data. The correct link is https://github.com/mlcclab/PyRAI2MD_publications/tree/main/Pentacene_dimers.

Response to referees

Our answers are in blue, and changes are in red.

Reviewer #1 (Remarks to the Author):

After the extensive clarifications and modifications provided by the authors, the manuscript is suitable for publication.

- *Answer:* Thanks for your time helping with the manuscript!

Reviewer #3 (Remarks to the Author):

The work "Machine learning photodynamics decode multiple singlet fission channels in pentacene crystal" by Li et. al. describes a novel and interesting way to simulate the photodynamics of a pentacene crystal using trajectory surface hopping with a multiscalar cluster approach. To speed up the computations and to enable trajectory surface hopping simulations in a reasonable time frame, neural networks were employed to replace the (expensive) CASSCF calculations.

The resulting decay rates are in excellent agreement with the experimental values for both the herringbone and the parallel dimer. The manuscript is generally well written and in the new version of the paper most of my initial questions and comments were answered. Still I have a few (new) open questions/comments that should be addressed before publication.

- *Answer:* Thanks for the summary of our manuscript and constructive comments!

1. The authors clarified that they selected SA6-CASSCF(4,4)/cc-pvdz as the method of choice, that shows a good compromise of efficiency and accuracy for the NN data. However, I do not understand the argument how SA6-CASSCF(4,4) was chosen. In the benchmark study SA6-CASSCF(4,4) is compared with SA6-CASSCF(8,8), XMS6-CASPT2(4,4) and MRSF-TDDFT. On page 4 the authors write: "Our benchmarks showed that the excited-state potential energy curves at the SA6-CASSCF(4,4)/cc-pVDZ level of theory are consistent with those obtained using XMS-CASPT2(4,4)/cc-pVDZ". But in both the answer to the reviewer as well as in the SI, the authors choose MRSF-TDDFT as the reference for the benchmark, because XMS6-CASPT2(4,4) "shows inconsistent results suggest that the XMS6-CASPT2(4,4)/cc-pVDZ method cannot capture all essential dynamical correlations". If this is the case, why do the authors emphasize that the SA6-CASSCF(4,4) is consistent with the XMS6-CASPT2(4,4) results (which I disagree, as it shows a different behavior for the two most important excited states)?

- *Answer:* Thank you for pointing out our unclear arguments regarding the choice of method. We agree that the SA6-CASSCF(4,4) results are not identical to the XMS6-CASPT2(4,4) results. The main difference is the positions of the S_0 and S_1 local minima in Figure 5 and Figure S3. The SA6-CASSCF(4,4)/cc-pVDZ calculations show S_0 minima at 6.1 Å and 4.0 Å for the herringbone and parallel dimers; the XMS6-

CASPT2(4,4)/ANO-S-VDZP results show S_0 minima at the 5.6 Å and 3.5 Å. The 0.5 Å differences in the intermolecular distances at the equilibrium structures do not significantly affect the predicted singlet-fission (SF) time constants because the molecular vibrations (e.g., Wigner sampling) lead to more than 1.0 Å variations in the intermolecular distances according to the trajectories in Figure 3e, 3i, 3g, and 3k. The S_2/S_1 gap is essential to the SF time constants. The SA6-CASSCF(4,4) results showed small S_2/S_1 gaps. The XMS6-CASPT2(4,4) results showed a slightly larger S_2/S_1 gap (0.65 eV) in the herringbone dimer with $R1 = 5.1$ Å than the SA6-CASSCF(4,4) results (0.34 eV). We do not expect this deviation to largely affect the SF time constants because the rest of the potential energy surface covered by the Wigner sampling (5.3–6.3 Å) shows sufficiently small S_2/S_1 gaps (< 0.6 eV) for SF.

We computed the S_0-S_1 and S_0-S_2 electron density differences in the herringbone dimers to examine the consistency between the SA6-CASSCF(4,4) and XMS6-CASPT2(4,4) calculations. We also included the MRSF-TDDFT results as they show excellent agreement in local minimum positions and S_2/S_1 gaps with SA6-CASSCF(4,4) results. The density difference $\Delta\rho$ is defined by the sum of the squared norm of the wavefunction weighted by the overall changes of the orbital populations, derived from the approach in *J. Chem. Phys.* **2024**, 161, 082503.

$$\Delta\rho_i = \sum_j w_{i,j} \sum_k (N_{i,j,k} - N_{0,j,k}) |\phi_k|^2$$

The subscripts i , j , and k denote the index of excited states, electronic configurations, and molecular orbitals. w is the weight of the configuration, N is the orbital occupation number, and ϕ is the molecular orbital. Blue and green refer to the electrons' depletion and accumulation.

The SA6-CASSCF(4,4) results in panel (a) show charge-transfer (CT) characters in S_1 and double excitation (DE) characters in S_2 at $R1 = 4.8 \text{ \AA}$. When $R1$ elongates, S_1 shows mixed CT and DE characters. We noted that S_2 becomes local excitation (LE) at $R1 > 6.1 \text{ \AA}$. Panel (b) shows the same S_1 nature at the XMS6-CASPT2(4,4) level as that at the SA6-CASSCF(4,4) level. The S_2 maintained the DE character when $R1 > 6.1 \text{ \AA}$ at the XMS6-CASPT2(4,4) level, suggesting an underestimated DE character at the SA6-CASSCF(4,4) level due to the lack of dynamical electron correlation. Nevertheless, the agreement in the S_1 nature ensures the correct electronic configuration analysis, as shown in Figure 5. The MRSF-TDDFT results in panel (c) are comparable to the XMS6-CASPT2(4,4) results when $R1 < 6.1 \text{ \AA}$. It overestimates the CT character in S_1 and underestimates the DE character in S_2 when $R1 > 6.1 \text{ \AA}$. This problem is caused by the incomplete double configurations generated by flipping a spin of one electron in a triplet reference. Therefore, we only considered the MRSF-TDDFT results as a supplementary reference of the PES and S_2/S_1 gaps, but not for electronic structure analysis.

The plots below illustrate the S_0 - S_1 and S_0 - S_2 electron density differences in the parallel dimers computed with the SA6-CASSCF(4,4), XMS6-CASPT2(4,4), and MRSF-TDDFT methods.

(a) SA6-CASSCF(4,4)/cc-pVDZ

(b) XMS6-CASPT2(4,4)/ANO-S-VDZP

(c) MRSF-TDDFT(PBE0)/cc-pVDZ

The SA6-CASSCF(4,4) and XMS6-CASPT2(4,4) results in panels (a) and (b) show good agreement in the mixed CT and DE characters in the S_0 - S_1 and S_0 - S_2 electron density differences. The MRSF-TDDFT calculations show consistent results when $R_2 < 4.7 \text{ \AA}$ in panel (c). As the distance increases, the missing double configurations underestimated the DE character in S_1 and an overestimated CT character in S_2 .

Collectively, our benchmarks show that SA6-CASSCF(4,4) calculations produced consistent S_1 nature with the XMS6-CASPT2(4,4) results and consistent PES and S_2/S_1 with the MRSF-TDDFT results. Given the excellent agreement between the predicted and experimental SF time constants, the SA6-CASSCF(4,4) is an appropriate method for studying the SF mechanism of pentacene dimers in crystals.

We added the above results in the Supporting Information Section S2, Figures S4 and S5. We also revised the main text to include the above important information, which reads, “Our benchmarks showed that the SA6-CASSCF(4,4)/cc-pVDZ method produced a consistent electronic structure description with XMS6-CASPT2(4,4)/ANO-S-VDZP results and the excited state potential energies are in line with the mixed-reference spin-flip (MRSF)-TDDFT⁴⁹ results (Figure S3-S5), in agreement with previous studies” on Page 4.

Following up on this, if MRSF-TDDFT was chosen as the reference for the benchmark and given that its computational costs is roughly half that of SA6-CASSCF(4,4), why was it not chosen as the method of choice?

- *Answer.* Thank you for suggesting using the MRSF-TDDFT method to compute the potential energy of the pentacene dimer. However, the highly efficient MRSF-TDDFT code was released several months after we finished the training data calculations and ML-photodynamics simulations.

The MRSF-TDDFT method was first released in GAMESS in mid-2022. Due to the limited efficiency and complexity of the contemporary version of GAMESS, we switched to the highly paralleled CASSCF methods in BAGEL when we started this work in early 2023. We completed our training data calculations and ML photodynamics simulations in mid-2023. When we had almost finished the manuscript, we heard the news about the development of OpenQP to improve the efficiency of MRSF-TDDFT. But at that time, OpenQP only had a collection of Fortran modules without a user interface for energy and gradient calculations. One of our co-authors, Jingbai Li, joined the development team in early 2024 and helped develop the OpenQP interface for flexible control of the energy and gradient calculations. The code was not released, and the related work was not published during the last revision of our manuscript. Thus, we were only allowed to include those results for benchmark purposes and to support the reliability of the SA6-CASSCF(4,4)/cc-pVDZ calculations. The OpenQP has now been formally released and published to J. Chem. Theory Comput. 2024, 20, 9464–9477 when we received the new comments, so we updated our reference in the main text (Ref 49) and Supporting Information (Ref S12).

Since the OpenQP is a newly released program, the MRSF-TDDFT calculations do not support electrostatic embedding, which requires further development. Electrostatic embedding is important in our ONIOM approach to describe electrostatic and polarization in the excitation of pentacene dimers. Thus, our training data calculations used the SA6-CASSCF(4,4) method with the electrostatic embedding of the surrounding point charges.

In addition, as we explained in our answer to your previous comments, the current MRSF-TDDFT formalism still misses some double configurations when the ROHF triplet reference state is an intermolecular charge transfer state. For pentacene dimers, the reference state is delocalized when the intermolecular distances are not significantly long, and our calculations of the vertical excitation energies at the

equilibrium structures (1.79 eV and 1.89 eV for the herringbone and parallel dimers) show good agreement with experiments (1.83 eV). Thus, the missing double configurations did not introduce significant problems in the excited state PES. When the intermolecular distances become substantially long, the missing double configurations result in inaccurate contributions from the CT and DE characters, even though the excitation energy seems fine. We are now working with the developer team to recover the missing double configurations in MRSF-TDDFT, which will allow us to rigorously apply this method to study the excited states of the dimeric system in the future.

The authors state in the SI that "The efficiency of the MRSF-TDDFT(PBE0)/cc-pVDZ calculations is realized by the newly designed quantum chemistry code, OpenQP, which is still under development and is only available for benchmark purposes while we finish this work". However, I do not fully understand this statement. What do the authors mean by "is only available for benchmark purposes"? Was the method only available, after the molecular dynamics simulations were performed?

- *Answer:* Thank you for the opportunity to clarify. We meant to write that the OpenQP code was not yet publicly available when we wrote the manuscript. It has since been released and published by J. Chem. Theory Comput. 2024, 20, 9464–9477 when we received the new comments. We found the current MRSF-TDDFT method still has a few limitations: 1) the electrostatic embedding is not available, and 2) some double configurations are missing in the spin-flip operation, leading to accurate multiconfigurational interactions when dimers separate at long distances. Thus, we did not use the MRSF-TDDFT method to train the NNs.

We have clarified our arguments and updated Ref S12 with the latest citation, which reads, "The efficiency of the MRSF-TDDFT(PBE0)/cc-pVDZ calculations is realized by the newly designed quantum chemistry code, OpenQP,¹² which was not released but only available for benchmark purposes while we wrote the manuscript. The OpenQP has now been formally published after the revision of our manuscript. The electrostatic embedding version of MRSF-TDDFT is still under development. Overall, the SA6-CASSCF(4,4)/cc-pVDZ shows a good balance between accuracy and efficiency" on Page S7.

2. The authors compute the training data for the ML using an electrostatic embedding approach, where one pentacene dimer is the QM region and all the others are included as point charges. The obtained NN model is used in an ONIOM approach, where the low level method is GNF2-xTB (GNF-FF). In ONIOM the interaction between the active side (here the pentacene dimer/NN) and the environment (here the other pentacene molecules) is computed at the low level method. From this it seems to me that the electrostatic interaction between the active side and the environment is (in parts) counted twice. Is this accounted for in the ONIOM scheme in combination with ML/NNs?

- **Answer:** Herein, we use an electrostatic embedding ONIOM scheme as implemented in fromage (*J. Chem. Theory. Comput.*, 2019, 15, 4, 2504-2516 and *J. Comput. Chem.*, 2020, 41, 10, 1045-1058) following the Morokuma's definition (*Chem. Rev.* 2015, 115, 12, 5678–5796). The energies are calculated by considering the following equation:

$$E_{total} = E_{GFN2-xTB,model} - E_{GFN2-xTB,dimer}^{EE} + E_{QM,dimer}^{EE}$$

The $E_{GFN2-xTB,dimer}^{EE}$, and $E_{QM,dimer}^{EE}$ terms correspond to the electrostatic embedding GFN2-xTB and QM energies of the pentacene dimer, respectively. Both include the one-electron interactions with the background charges in their electronic Hamiltonian, as shown in the following equation:

$$E_{GFN2-xTB,dimer}^{EE} = E_{GFN2-xTB,dimer} - \sum_J \sum_N \frac{q_J q_N}{R_{I,N}} + \sum_J \sum_N \frac{Z_J q_N}{R_{J,N}}$$

$$E_{QM,dimer}^{EE} = E_{QM,dimer} - \sum_J \sum_N \frac{q_J q_N}{R_{I,N}} + \sum_J \sum_N \frac{Z_J q_N}{R_{J,N}}$$

N, J, and I denote the environment atom, dimer atoms, and dimer electrons, respectively, and q_N is the embedded charge for the environment atom N; q_J is the charge of one electron of the dimer atom J, equal to 1 in the atomic unit. Z_J is the nuclear charge of the dimer atom J. $R_{I,N}$ is the distance between the dimer electron I and the environment atom N, and $R_{J,N}$ is the distance between the dimer atom J and the environment atom N. We used the RESP charges of the environment atoms computed with the ω B97XD/def2-TZVP method as implemented in the Gaussian16 program.

Subtracting $E_{GFN2-xTB,dimer}^{EE}$ from $E_{GFN2-xTB,model}$ removes the ground-state electronic interactions between the dimer and environment, and by adding $E_{QM,dimer}^{EE}$ recovers the electronic interactions between the excited state of the dimer and the environment. Thus, this procedure avoids double-counting of the electronic interactions.

The NNs learn the $E_{QM,dimer}^{EE}$ term for ML-photodynamic simulations, where all training data were computed at the SA6-CASSCF(4,4)/cc-pVDZ level of theory with the electronic embedding of RESP charges as described above. As such, the NNs implicitly learn the electronic interactions with the environment in the $E_{NN,dimer}^{EE}$ term. During the ML-photodynamics simulations, the $E_{GFN2-xTB,model}$ and $E_{GFN2-xTB,dimer}^{EE}$ terms were calculated on-the-fly, where the $E_{GFN2-xTB,dimer}^{EE}$ term includes the electronic embedding of the RESP charges. Therefore, the subtraction of $E_{GFN2-xTB,dimer}^{EE}$ and addition of $E_{NN,dimer}^{EE}$ (learned from $E_{QM,dimer}^{EE}$) to $E_{GFN2-xTB,model}$ does not double count the electrostatic interactions twice in the total energy. We have expanded the Computational Details to clarify these points, which reads, “The $E_{GFN2-xTB,dimer}^{EE}$, and $E_{QM,dimer}^{EE}$ terms correspond to the electrostatic embedded GFN2-xTB and QM energies of the pentacene dimer, respectively” and “In our ML-photodynamics simulations, the QM calculations were replaced by NN predictions, where all the

training data were computed with the electrostatic embedding from the RESP charges” on Page 11-12.

3. The authors write at several points that they use SA6-CASCI(4,4) instead of SA6-CASSCF(4,4). Is the same method meant? If yes, please use SA6-CASSCF in all places, if not why was SA6-CASCI used and how does it perform in comparison?

- *Answer:* SA6-CASSCF (4,4) and SA6-CASCI (4,4) are technically different. As explained in this reference (*J. Chem. Phys.* 154, 2021, 090902), the SA6-CASSCF (4,4) calculations optimize the active orbitals and configuration interaction (CI) coefficients simultaneously, while the SA6-CASCI(4,4) calculations only optimize the CI coefficient given a set of pre-determined active orbitals. Our work first computed the active orbitals with the SA6-CASSCF(4,4) calculations. Then, we localized the active orbitals with the Pipek-Mezey (PM) method for the SA6-CASCI(4,4) calculations. Since the PM orbital localization is a unitary transformation, the SA6-CASCI(4,4) calculations produced the same energy as the SA6-CASSCF(4,4) results. The difference between SA6-CASCI(4,4) and SA6-CASSCF(4,4) is in the CI coefficients, i.e., the contributions of the configurations, as they are computed with different active orbitals.

We performed the SA6-CASCI(4,4) calculations because the SA6-CASSCF(4,4) calculations produced delocalized π -orbitals on both pentacene monomers, as shown in Figure S2, which is not straightforward for analyzing the electronic configuration of the coupled triplet-triplet pair (TT) configuration. The localized active orbitals in Figure 1c present a more intuitive chemical meaning than the delocalized orbitals, as each monomer carries one electron. As such, the CI coefficients in the SA6-CASCI(4,4) calculations directly reflect the contributions of the TT configuration in the excited states.

In the first paragraph of page 10, we discussed the reason for choosing SA6-CASCI(4,4) in our electronic configuration analysis. Because SA6-CASCI(4,4) produced the same energy and forces as SA6-CASSCF(4,4) in our work, we only used SA6-CASSCF(4,4) to compute the training data.

Minor remarks:

page 6: "The predicted SF time constants also match with the decay time constants (35 and 75 fs)". I assume that is a typo and it should be 78 fs.

- *Answer:* You are correct. We have corrected this typo to 78 fs accordingly.

SI:

page 1: "electron-hele distances"

- *Answer:* Thank you for pointing out the typo. We have corrected it accordingly.